



# Use of absorption optical indices to assess seasonal variability of dissolved organic matter in amazon floodplain lakes

Maria Paula da Silva[1]*, Lino A. Sander de Carvalho[2], Evlyn Novo[1], Daniel S. F. Jorge[1], Claudio C. F. Barbosa[2]

[1] Remote Sensing Division, National Institute for Space Research, 12227-010, São José dos Campos, Brazil
[2] Image Processing Division, National Institute for Space Research, 12227-010, São José dos Campos, Brazil;
*Correspondence to*: Maria Paula da Silva (maria-paula.da-silva@ufz.de)

**Abstract.** Given the importance of DOM in the carbon cycling of aquatic ecosystems, information on its seasonal variability is crucial. This study assesses the use of available absorption optical indices based on in situ data to both characterize the seasonal variability of the DOM dynamics in a highly complex environment and test their viability of being used for satellite remote sensing on large scale studies. The study area comprises four lakes located at the Mamirauá Sustainable Development Reserve (MSDR). Samples for the determination of coloured dissolved organic matter (CDOM) and remote sensing reflectance

(Rrs) were acquired in situ. The Rrs was applied to simulate MSI visible bands and used in the proposed models. Differences between lakes were tested regarding CDOM indices. Significant difference in the average of $a_{CDOM}$ (440), $a_{CDOM}$ spectra and $S_{275-295}$ were found between lakes located inside the flood forest and those near the river bank. The proposed model showed that $a_{CDOM}$ can be used as proxy of $S_{275-295}$ during rising water with good validation results, demonstrating the potential of Sentinel/MSI imagery data in large scale studies on the dynamics of DOM.

## 1 Introduction

Floodplain is a type of wetland characterized by a mosaic of landscapes which oscillates periodically between aquatic and terrestrial systems. This oscillation represents a key aspect in the biogeochemistry, ecology and hydrology of floodplain lakes (Junk et al., 1989; Moreira-Turq et al., 2004). Among other effects, the flood pulse (*sensu* Junk et al., 1989) affects the proportion of autochthonous and allochthonous sources contributing to the dissolved organic matter (DOM) pool in floodplain

lakes throughout the year (Melo et al., 2019).

DOM represents the largest pool of organic carbon in the aquatic environment (Cole et al., 2007; Tranvik et al., 2009) and it has an important role in the ecosystem carbon budgets (Dalmagro et al., 2019; Pangala et al., 2017; Richey et al., 2002). Besides that, DOM also control light attenuation and availability (Coble, 2007; Kirk, 2011), playing a vital role in primary productivity of aquatic ecosystems and consequently fisheries and other food webs (Maia and Volpato, 2013; Volpato et al.,

30  2004).

DOM concentration in the environment is usually determined by the concentration of dissolved organic carbon (DOC) (Toming et al., 2016). However, simple measurement of DOC concentration can restrict the study of the seasonal variation in the DOM





composition and origin because it is only related to DOM bulk (Jaffé et al., 2008). In these cases, information regarding DOM quality such as ultraviolet (UV) and visible absorption measures, as well as fluorescence, can aid in the explanation of DOM

variability (Li and Hur, 2017). Helms et al., (2008) have shown that the spectral slope calculated in the range of 275 and 295 nm ($S_{275-295}$) is an indicator of DOM molecular weight and a tracer of degradation processes, being a relevant parameter in the study of DOM dynamic. Based on measurements of CDOM and DOC, a study made on Mississippi and Atchafalaya River revealed that the rates of water exchange between river and floodplain increase DOC concentrations and alter DOM composition (Shen et al., 2012). Also based on the optical properties of DOM, Spencer et al. (2008) observed higher values of

CDOM during the spring flush at Yukon River basin, when DOM had a remarkably high content of aromatic vascular plant material derived from surface soil and litter layers.

In order to study DOM dynamics in a large spatial-temporal scale satellite images have been assessed as a source of CDOM optical information. In a recent Pan-Arctic study, Fichot et al. (2013) showed that $S_{275-295}$ can be directly estimated from satellite images using a multi-linear parameterization of MODIS marine reflectance. However, the relationship proposed by Fichot et

al. (2013) between Remote Sensing Reflectance (Rrs) and $S_{275-295}$ is based into two assumptions: CDOM optical dominance in water and co-variation between CDOM and other particulate matter. To circumvent these assumptions, Vantrepotte et al. (2015) investigating the relationship between CDOM and DOC showed that $S_{275-295}$ can be estimated from MODIS data using $a_{CDOM}$ (412) as proxy. Nonetheless, both studies (Fichot et al., 2013; Vantrepotte et al., 2015) used MODIS data whose spatial resolution (250-1000 m) restricts the application to inland water studies. Conversely, the availability of Multi-Spectral

Instrument (MSI) images, on board of the Sentinel-2A (June/2015) and Setinel-2B (March/2017), have expanded the potential of remote sensing application for DOM monitoring. Its high spatial (10 and 20 m), temporal (5 days) and radiometric (12-bit) resolutions increase the viability of monitoring seasonal changes in $a_{CDOM}$ in inland waters (Toming et al., 2016).

The main objectives of this study are to: i) investigate the variability of $a_{CDOM}$ in floodplain lakes during the receding and rising phases of the Solimões River, ii) compute $S_{275-295}$ to examine its potential for distinguishing differences in DOM by comparing

them in those two hydrograph phases; and iii) propose an algorithm to estimate $a_{CDOM}$ (440) as a proxy for $S_{275-295}$ to support future application of remote sensing data for DOM studies.

## 2 Material and methods

### 2.1 Study area

The study sites are four lakes located in the floodplain built at the confluence between Solimões and Japurá rivers, near Tefé

and inside the Mamirauá Sustainable Development Reserve (MSDR) (Figure 1b), a well-preserved flooded forest and under low human pressure (Ayres, 1995; Castello et al., 2009; Mori et al., 2019; Queiroz, 2007).

The seasonal flood is caused by both, the upper Amazon basin and local rainfall during the rainy season (from December to May, with an average of 300 mm/month) and by the annual melt of the Andean cordillera during the austral summer (Junk, 1989). The yearly MSDR flood pulse causes (Queiroz, 2007), in average, 12 meters amplitude in the water level, between the





dry (September to November) and the flood season (May and mid-July). The rising of the water level begins in January and goes up to late April while the water receding starts in July. During the flood period, which begins in May, the floodplain is totally occupied by water until the beginning of the receding phase (Affonso et al., 2011).

The lakes were selected according to criteria defined in Jorge et al. (2017a) to guarantee access to them throughout the hydrological year and sizes compatible to the spatial resolution of satellite sensors (MSI/ Sentinel-2A and OLI/ Landsat 8).

Additionally, the lakes have intrinsic differences: two of them (Buabuá and Mamirauá) are small perennial lakes surrounded by flood forest, isolated, while the others (Pantaleão and Pirarara) are lakes connected to the Japurá river along the entire hydrological year, with variable size and depth in response to the flood pulse.

## 2.2 Data source

Data were acquired in Buabuá, Mamirauá, Pantaleão and Pirarara lakes by the Instrumentation Laboratory for Aquatic Systems

team (LabISA – http://www.dpi.inpe.br/labisa) of the National Institute for Space Research (INPE). Details of the fieldwork and DOC concentration measurement are provided in Jorge et al. (2017a, 2017b).

The field campaigns were carried out in March-April and July-August of 2016 which corresponds to the rising and receding water level of Solimões River. Table 1 contains the sampling points and the DOC concentration measured, in total 87 stations were sampled distributed among the lakes and seasons.

## 2.3 Measurements

### 2.3.1 Remote sensing reflectance

The radiometric measurements were performed using three inter-calibrated RAMSES TriOS sensors. These sensors simultaneously acquire water leaving radiance (Lw), sky radiance (Lsky) and down welling irradiance (Ed) which were used to estimate the Rrs according to Mobley (1999) as described in Jorge et al. (2017b). The sensors operate in the range between

350 and 900 nm.

In situ Rrs was used to simulate the reflectance of the MSI bands (https://earth.esa.int). For this, MSI Relative Spectral Response (RSR) of the sensor was used (Equation 1):

$$R_{rs}(B_i) = \frac{\int_{\lambda_m}^{\lambda_n} RSR(\lambda) \cdot R_{rs\,m(\lambda)} d\lambda}{\int_{\lambda_m}^{\lambda_n} RSR(\lambda) d\lambda}, \qquad (1)$$

where Rrs_m is the Rrs measured in situ and Rrs (Bi) is the Rrs simulated for the i-th band of Sentinel-2A, in the wavelength

range of λm to λn. MSI RSR were taken from the user guide of the sensor.

### 2.3.2 CDOM Absorption Coefficient

Water samples were filtered first through Whatman GF/F (0.7 μm) filters (burned at 400 ºC) and then through 0.22 μm pore size polycarbonate filter. The filtrated sample was stored in sterilized dark glass bottles and kept refrigerated until analysis.





During the analysis, all samples were kept at ambient temperature. CDOM spectral absorbance was measured with a Shimadzu
UV-2600 spectrophotometer in the wavelength range between 220 and 800 nm, with increments of 1 nm and converted to
$a_{CDOM}(\lambda)$ according to Equation (2) (Bricaud et al., 1981):

$$a_{cdom}(\lambda) = \frac{2,303 \cdot A(\lambda)}{L} \; , \hspace{4cm} (2)$$

where $A(\lambda)$ is the spectral absorbance of the filtered sample in the specific wavelength $\lambda$ (nm) and L is the cuvette path length
(0.1 m).

The average of $a_{CDOM}$ between 750 and 800 nm was used to correct the residual absorption spectra due to baseline drift,
temperature, scattering, and refractive effects (Coble, 2007).

### 2.3.2 Spectral slope determination

The spectral slope in the interval between 275 and 295 nm ($S_{275-295}$) was computed according to the Equation 3 (Bricaud et al.,
1981), using non-linear fit (Helms et al., 2008). This function describes the behaviour of $a_{CDOM}(\lambda)$ along the electromagnetic
spectrum and is expressed as:

$$a_{cdom}(\lambda) = a_{cdom}(\lambda_{ref}) \cdot e^{-S(\lambda-\lambda_0)} \; , \hspace{3cm} (3)$$

where S is the spectral slope parameter ($nm^{-1}$) between the wavelength interval of $\lambda - \lambda_0$ and $\lambda_{ref}$ is a reference wavelength
(nm).

### 2.3.2 Statistical analyses

The temporal variability of the $a_{CDOM}(440)$ among the lakes was analysed using box-plot and scatter plot. This wavelength
was chosen due to the high CDOM absorption at this wavelength (Jorge et al., 2017a), being also a region used as reference in
remote sensing studies, at least, in the last thirty-six years (Bricaud et al., 1981; Brezonik et al., 2015; Bukata et al.,1995;
Werdell et al., 2018).

The coefficient of variance (CV) was also computed for assessing $a_{CDOM}(440)$ variability. Kruskal Wallis test (significance
level of 95%) was applied to test the existence of significant differences between lakes and hydrograph phases regarding $a_{CDOM}$
(440) values as follows: i) in a first run, the test used the entire data set; ii) in a second run, in attempt to test the influence of
Buabuá and Mamirauá samples acquired during the rising period, those samples were removed.

The mean $S_{275-295}$ among months of the same hydrograph phases was computed in an attempt to analyse their variability within
each lake and phase. Scatter plot between $a_{CDOM}(440)$ and $S_{275-295}$ were also built for different hydrograph phases to assess
eventual relationships and define the approach to develop the remote sensing model.

### 2.3.2 Model calibration and validation

The model proposed by Vantrepotte et al. (2015) and a power function (Equation 4) were tested.



$$S_{275-295} = x \cdot a_{cdom}(440)^{-y} \, , \tag{4}$$

where x and y are fitting coefficients of the Equation (4).

Augusto-Silva et al. (2014) proposed the use of Monte Carlo simulation to overcome the limited number of sampling stations for computing the coefficients. Thus Monte Carlo was applied to Equation 4 to find the coefficients and validate the model. Out of 42 $S_{275-295}$ samples collected during the rising phase, 29 were randomly selected for model calibration. The 13 remaining samples were used to validate the model. This process was repeated 104 times and the Mean Square Error (MSE) and equation coefficients (x and y) were recorded, at each iteration.

Final model selection (most representative model based on MSE modal value) follows Augusto-Silva et al. (2014) procedure: i) constructing a histogram of MSE; ii) computing of mean and standard deviations of model's coefficients in the most frequent error interval; iii) ranking of coefficients in the range of mean ± standard deviation according to their MSE, iv) selecting the model with the smallest MSE.

The chosen model was validated using the 13 remaining samples (not used in the calibration process) and the final accuracy

was assessed following the metrics: coefficient of determination ($r^2$), MSE and normalized root mean square error in percentage (%NRMSE).

Once the relationship between $S_{275-29}$ and $a_{CDOM}$ (440) was modelled, another algorithm was calibrated and validated to estimate $a_{CDOM}$ (440) using the simulated MSI Rrs. To the authors' knowledge, there is no model developed for environments with the optical specificity of the Amazon floodplain lakes characterized by both high CDOM and Non-Algal Particle (NAP)

contribution to the Rrs. Therefore, a new model for estimating $a_{CDOM}$ (440) is proposed. Besides the usual exponential band ratio in the visible region, a ratio between near infrared bands was introduced to remove NAP contribution from its inorganic fraction. The rational for introducing this ratio is the null signal of CDOM and the dominance of NAP in this wavelength range (Kirk, 2011). Previous studies also have shown that the inclusion of bands at wavelengths >600 nm increases the accuracy of the CDOM estimation model (Chen et al., 2017; Zhu et al., 2014). Thus, to determine $a_{CDOM}$ (440), the exponential of the ratio

between bands 6 (λcentral wavelength =740 nm) and 5 (λcw =705 nm) are subtracted from the exponential of the ratio between bands 2 (λcw =490 nm) and 3 (λcw =560 nm) as showed in Equation 7:

$$a_{cdom}(440) = x \cdot e^{(B2/B3)} - (y \cdot e^{(B6/B5)} + z) \, , \tag{5}$$

where, x, y and z are the coefficients. B2, B3, B5 and B6 are the MSI sensor simulated bands 2, 3, 5 and 6.

Monte Carlo simulation was similarly performed to select the most representative model for estimating $a_{CDOM}$ (440) as a

function of Rrs. The validation process also followed the same procedure previously described for the slope.



## 3 Results

### 3.1 Seasonal and spatial variability of CDOM

The highest amplitude of $a_{CDOM}$ (440) in the entire data set occurred in March (1.22 to 5.46 m$^{-1}$) and April (1.60 to 5.97 m$^{-1}$), with averages of 2.56 and 3.01 m$^{-1}$, respectively. In July and August, the amplitude was smaller (1.32 to 2.03 m$^{-1}$ and 1.27 to

2.19 m$^{-1}$, respectively) and both averaged below 2 m$^{-1}$. No spatial variation was observed in $a_{CDOM}$ (440) within lake.

The water level was almost the same along the sampling campaigns (30.04 ± 1.38 m). High variability (CV=52.45%) and magnitude of $a_{CDOM}$ (440) occurred at the rising phase, while in the receding, both, $a_{CDOM}$ (440) magnitude and variability (CV=14.74%) were much smaller. Additionally, $a_{CDOM}$ (440) displayed different values among lakes in March and April, while in July and August, the values had higher similarity (Figure 3).

Kruskal Wallis results using samples from all lakes and dates indicated significant differences in $a_{CDOM}$ (440) between lakes and hydrograph phases (p= 7.3101e$^{-06}$). After the removal of Buabuá and Mamirauá samples acquired in March and April (rising), Kruskal Wallis results showed no significant differences in $a_{CDOM}$ (440) values (p=0.51). This indicates that DOM at Buabuá and Mamirauá, during the rising phase have a much higher absorption at 440 nm than those of the remaining lakes and months.

### 3.2 CDOM absorption spectra

The entire set of $a_{CDOM}$ spectra (Figure 4) can be divided in two groups. The first group comprises Buabuá and Mamirauá spectra acquired at the rising phase, with $a_{CDOM}$ at 254 nm ranging between 65 and 95 m$^{-1}$. The second group is composed by the Pantaleão and Pirarara spectra at rising phase and the samples of all the lakes acquired during the receding phase, with $a_{CDOM}$ at 254 nm ranging from 26 to 35 m$^{-1}$. It is also noticeable a feature with a shape of a shoulder between 245 and 290 nm

in the absorption spectra (black arrow in Figure 4). Thus, during the rising phase there are differences between the spectra collected in the lakes surrounded by the flooded forest and those near the river. During the receding phase, however, this difference no longer exists, and all spectra have lower $a_{CDOM}$ (254) values.

The analysis of average $S_{275-295}$ of each hydrograph phase also indicates the existence of differences between both, phases and lakes (Figure 5). The scatter plot displays the presence of two distinct groups: one including Mamirauá and Buabuá samples,

and the other, Pantaleão and Pirarara. During the rising phase, Pantaleão and Pirarara have higher $S_{275-295}$ than Buabuá and Mamirauá. However, during the receding phase, the highest $S_{275-295}$ occurred at Buabuá and Mamirauá.

$S_{275-295}$ in all samples from Buabuá and Mamirauá are near or under 0.015 nm$^{-1}$ in rising phase and equal or higher than 0.016 nm$^{-1}$ in the receding phase. However, $S_{275-295}$ in all Pantaleão and Pirarara samples are above 0.015 nm$^{-1}$ in the rising phase and below 0.0155 nm$^{-1}$ in the receding phase, except for one single sample from Pantaleão.


The spectral slope ratio ($S_R$) between the wavelength intervals of 275-295 nm on 350-400 nm was applied to trace DOM sources in a tropical river-wetland system (Dalmagro et al., 2017). This ratio was examined in the current study and shows the





same pattern observed in $S_{275-295}$, indicating differences between the lakes surrounded by flooded forest and those near the river (see supplementary material S.1). The high relationship between $S_{275-295}$ and $S_R$ indicates that these parameters can be

tracking similar pools of DOM (Hansen et al., 2016). Since $S_{275-295}$ also has been preferred in remote sensing studies (Fichot et al., 2013; Vantrepotte et al., 2015), the model proposed in the present study used $S_{275-295}$ instead of $S_R$.

### 3.3 Seasonal relationship between $a_{CDOM}$ and $S_{275-295}$

The model proposed by Vantrepotte et al. (2016) was tested using the entire data set, but a power-law function provided a better fit (Figure 6).

The relationship between $a_{CDOM}$ (440) and $S_{275-295}$ varies between hydrograph phases (Figure 7). In the receding phase, a wide range of $S_{275-295}$ (0.014 to 0.0165 nm$^{-1}$) was observed for a narrow range of $a_{CDOM}$ (440) (between 1 and 2 m$^{-1}$), indicating no correlation between those variables (Figure 7a). However, in the rising phase, a power trend has been observed: as $a_{CDOM}$ (440) increases from 1 to 6 m$^{-1}$, $S_{275-295}$ decreases from 0.0165 to 0.0142 nm$^{-1}$ (Figure 7b). These results indicate that in the rising phase, $a_{CDOM}$ (440) can be used as proxy of $S_{275-295}$ since in the receding phase it remains almost constant. Therefore, the model

was developed just for samples acquired in the rising phase. The selected model shows a satisfactory fit (MSE<0.0001) and is described in Equation 6:

$$S_{275-295} = 0.016 \, a_{CDOM}(440)^{-0.064} \quad , \qquad\qquad (6)$$

Validation results were also satisfactory indicating the feasibility of estimating $S_{275-295}$ from $a_{CDOM}$ (440) (Figure 8a). However, the estimated $S_{275-295}$ diverges from the 1:1 line for values above ~0.015 nm$^{-1}$, indicating that the model is better parameterized

for values of $S_{275-295}$ smaller than 0.015 nm$^{-1}$.

Once a relationship between $a_{CDOM}$ (440) and $S_{275-295}$ was established for rising water, the model to estimate $a_{CDOM}$ (440) via Rrs was also tested for this period. The final model had satisfactory accuracy (MSE = 0.65 m$^{-1}$) and is expressed in Equation 7:

$$a_{CDOM}(440) = 4.39^{\frac{B_2}{B_3}} + 0.59^{\frac{B_6}{B_5}} - 6.67 \quad , \qquad\qquad (7)$$

Model validation (Figure 8b) shows that almost 70% of the estimated values are within the 95% confidence interval and the statistics parameters (r², MSE and %NRMSE) present good accuracy in the estimates of $a_{CDOM}$ (440).

### 4 Discussion

The variability of $a_{CDOM}$ (440), $a_{CDOM}$ spectra and $S_{275-295}$ along the hydrological year (Figures 1, 3, 4 and 5) seems to be related to the hydrograph phases and lake geographical location in the floodplain. As seen in Figure 1, lakes Mamirauá and Buabuá

are located in the middle of the floodplain, far from both main rivers, Solimões and Japurá and surrounded by High and Low Várzea Forests (Ferreira-Ferreira et al., 2015 Figure 5). While Pantaleão and Pirarara are lakes located near to Japurá River, subjected to both river inputs and Solimões River flood pulse.



The water level in the floodplain is quite similar between the rising and receding seasons, suggesting that the water level is not the major factor explaining the variability of those optical variables. The Solimões flood pulse phase is, therefore, the key variable controlling the variability of CDOM index. During the rising water level, the Solimões inflow into the floodplain as overland flow crosses a large area of forest and carries a considerable amount of organic matter in different stages of decomposition into Buabuá and Mamirauá lakes. Pantaleão and Pirarara lakes, however, are far from Solimões, being connected to Japurá River located in the eastern extreme of the floodplain; therefore they are not affected by the Solimões overland flow in the beginning of the rising phase, receiving a minor input of organic matter as Buabuá and Mamirauá (Table 1).

As the study area consists entirely of a floodplain that is subject to marked seasonal flooding (about 12 m), during the high water, the entire ecosystem is flooded (Affonso et al., 2011). According to Ferreira-Ferreira et al. (2015), the entire area seen in Figure 1 are flooded for periods of up to 295 days in a year depending on the flood peak. In this study, the high-water phase was not sampled considering that previous studies (Affonso et al., 2011) indicated that during the high water all water bodies become interconnected with the main channels and rivers displaying the lowest spatial variability in all limnological variables, including DOC concentration. Actually, DOC coefficient of variation among sampled water bodies dropped from 53.87 % in the low water to 20.89 % in the high-water of 2009 hydrological year (Affonso et al., 2011). Considering that in the Amazon basin, DOC accounts for 50% of total organic matter and that floodplain areas are relevant sources of DOC to the Solimões/Amazon River (Richey et al., 1990), in the present study we assume that it is possible that the differences in CDOM optical properties among Mamiraua/Buabuá and Pirarara/Pantaleão are related to the fact that the flood wave have not reached the eastern margin of the floodplain at the onset of the rising water phase.

In the rising phase, the water coming mainly from the Solimões river undergoes overbank flooding (Figure 1c), overtopping its channel and flowing across the litter through the forest before reaching the lakes (Junk, 1989), carrying a great amount of organic matter accumulated during the lower water season. As explained in the previously paragraph, in the beginning of the rising phase, the water from Solimões does not reach all the floodplain lakes at the same time. Therefore, in this period, DOM is expected to have significant differences between those lakes surrounded by flooded forests located near the Solimões River and those connected to Japurá River, located in the extreme eastern boundary of the study area. During the rising water phase, the water path to Pirarara and Pantaleão through the flooded forest is small, because they are closely connected to Japurá River. At that time, the Solimões overbank flood of the high-water season, responsible for homogenizing limnologic properties (Abdo and Silva, 2004; Almeida and Melo, 2009; Carvalho et al., 2001; Henderson, 1999; Queiroz, 2007) in floodplain lakes have not occurred yet.

According to previous research, high molecular height (HMW) DOM has a lower $S_{275-295}$ than low molecular height (LMW) DOM (Helms et al., 2008). Also, HMW can be an indicative of allochthonous DOM since it is composed of refractory compounds such as lignin and cellulose. Regarding the assessment of $S_{275-295}$ values in this study (Figure 5), differences were found between both, lakes and hydrograph phases. During the rising phase, Pantaleão and Pirarara have higher $S_{275-295}$ (>0.015 nm$^{-1}$) than those of Buabuá and Mamirauá (<0.015 nm$^{-1}$), suggesting that DOM at lakes near that river have lower molecular





weight than those surrounded by forest. These results agree with previous studies (Melo et al., 2019; Shen et al., 2012; Spencer et al., 2008) indicating the presence of HMW DOM during rising water. However, in the present study the authors do not have data to corroborate the optical analyses regarding the origin and molecular weight of DOM.

In our study, we didn't find high correlation between $a_{CDOM}$ (440) and $S_{275-295}$ for the data set including samples acquired in all hydrograph phases. During the receding water phase it is difficult to draw conclusions regarding DOM origin, since the DOM present in the lakes can be old and highly degraded (Wagner et al., 2019). During the rising phase, a higher correlation between $a_{CDOM}$ (440) and $S_{275-295}$ can be found. This means that high (low) $a_{CDOM}$ (440) values correspond to low (high) $S_{275-295}$ values, suggesting the presence of HMW (LMW) substances. In this way, it seems that $a_{CDOM}$ (440) and $S_{275-295}$ are optical absorption

indices that can be used to trace different CDOM dynamics between lakes and hydrograph phases in floodplain lakes. Since literature shows that these indices can be estimated via remote sensing data (Brezonik et al., 2015; Vantrepotte et al., 2015), we try to model their relationship with the remote sensing reflectance (Rrs). However, because of the differences in CDOM dynamic among hydrography phases, we only model the relationship between the variables for the data set sampled during rising water, which is a key moment when the floodplain receives large amount of water coming from different Amazon basin

habitats which washes the floodplain floor and carries large amount of organic matter accumulated along the hydrological year.

There are several models relating $a_{CDOM}$ (440) and remote sensing data in literature, as reported by Zhu et al. (2015), but since they are empirical models they are environmental and seasonal dependent. Kutser et al., (2016) tried to calibrate a model using data from Estonian lakes, Três Marias Reservoir (Brazil) and a floodplain lake located in Amazon (Curuai Lake). However,

they could not develop a fitting model describing the entire data set, showing that model development depends on DOM quality and degradation dynamics (Hansen et al., 2016). Models available in literature usually use the ratio between the green and red bands (Toming et al., 2016; Zhu et al., 2014). In our study, we tested the correlation between $a_{CDOM}$ (440) values and the ratio between the green and red bands, but we didn't find a significant correlation (see supplementary material S.2). Thus, we proposed a new model to estimate $a_{CDOM}$ (440), using additional bands (Equation 7).

Despite the small number of samples, this study shows that it is possible to estimate $S_{275-295}$ from $a_{CDOM}$ (440) during one crucial hydrograph phase (rising phase), notwithstanding their hydrodynamic differences. Both the MSE and %NRMSE (<0.0001 m$^{-1}$ and 9.40%) computed in this study are in the range of models available in the literature (Vantrepotte et al., 2015; Fichot et al., 2013), showing potential for estimating $S_{275-295}$ from $a_{CDOM}$ (440). Therefore, a model for $a_{CDOM}$ (440) estimation was also proposed. The $a_{CDOM}$ (440) model also provided MSE and %NRMSE (0.53 m$^{-1}$ and 15.12%) which is considered

good estimates based on remote sensing methods. Those modelling results, therefore, are encouraging suggesting that MSI images, when available, might be applied for studying DOM properties of the Amazon floodplain lakes during the rising. However, the models have limitations, which are: 1) its empirical nature demanding calibration for application in other datasets; and 2) the small range of $a_{CDOM}$ sampled (1.2 to 6.0 m$^{-1}$) and $S_{275-295}$ (0.0142 to 0.0165 nm$^{-1}$), indicating the need of new experiments including a larger number of lakes spread in a wider range of distance from the Solimões bank, a wider span

of the rising hydrograph phase and DOM molecular analyses in order to validate the optical indices.

## 5 Conclusions

The present study indicates that the use of the optical indices, $a_{CDOM}$ and $S_{275-295}$, provided a deeper understanding on the connections between Solimões and Japurá river flood pulse and DOM dynamics in the Amazon floodplain lakes.

The results in this study corroborates the findings in the most recent literature and indicates that there is an urgent need of research to explore new types of indices integrating both, optical spectral properties and remote sensing data. Also the results seem very promising concerning the costs and time consuming necessary for DOM analysis in laboratory, highlighting the potential of DOM studies from remote sensing.

The empirical model relating Rrs and $a_{CDOM}$ (440); $a_{CDOM}$ (440) and $S_{275-295}$ provided robust statistics indicating the high potential of MSI sensor for estimating $S_{275-295}$ during the rising water. Even though this study is the first attempt of using simulated MSI data to estimate $S_{275-295}$ in Amazon floodplain lakes, the results herein discussed seem very promising particularly considering the new generation of satellite-borne sensors with higher temporal resolution.

*Author contributions.* MPdaS, LASdeC, EN and CCFB planned and designed the research. DSFJ and CCFB carried out parts of the field work and conducted a first version of data processing. MPdaS did the statistical analysis and wrote the paper with contributions from all co-authors.

*Competing interests.* The authors declare that they have no conflict of interest.  10

*Acknowledgements.* This study was funded by São Paulo Research Foundation (FAPESP 2014/23903-9), National Council for Scientific and Technological Development (CNPq 461469/2014-6 and CNPq 304568/2014-7) and by the project "Environmental Monitoring by Satellite in the Amazon Biome" from the Brazilian development bank (MSA-BNDES 1022114003005). Da Silva was funded by the CNPq (131242/2016-4). We are very grateful to Vitor Martins, Renato Ferreira, Jean Farhat, Franciele Sarmento, and Waterloo Pereira Filho for their assistance during field missions. We would like to thank Dr. Helder Queiroz and the Mamirauá Institute for all the support. We also would like to thank Otávio Cristiano Montanher for the correction of flow rate data and Ivan Bergier for suggestions.

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





**Figure 1 – Study area. (a) Overview Amazon. (b) OLI/ Landsat 8 true color image from July 30th of 2015 showing the study area**
**and sample stations lakes: (I) Buabuá; (II) Mamirauá; (III) Pantaleão; and (IV) Pirarara. (c) Water flow rate at Japurá and Solimões**
**rivers (Brazilian Water National Agency - ANA).**



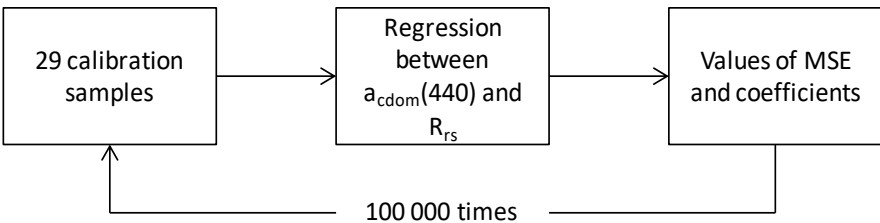

**Figure 2 – Diagram of Monte Carlo simulation.**


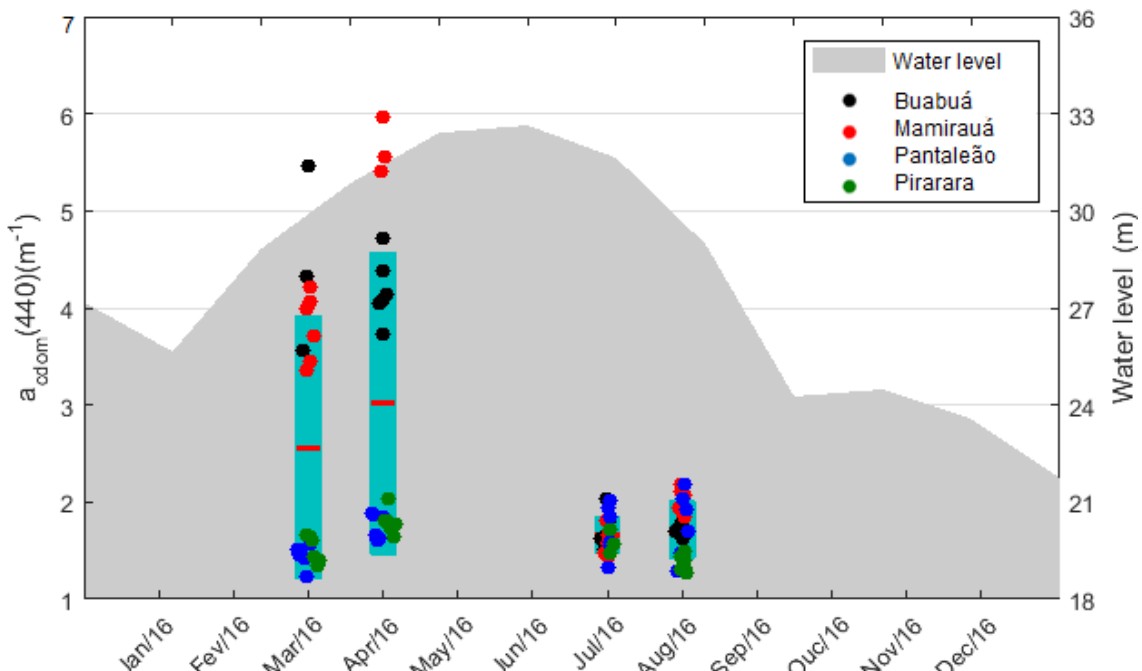

**Figure 3 – Seasonal box-plot of a$_{CDOM}$ (440) (m$^{-1}$) and water level (m) of Mamirauá channel in 2016. The red lines are the mean a$_{CDOM}$(440); each dot represent the a$_{CDOM}$ (440) value at each sample station of lakes; and the blue boxes represent the interval between the first and third quartile. Water level data were acquired from Mamirauá Sustainable Development Institute (MISD, 2017).**






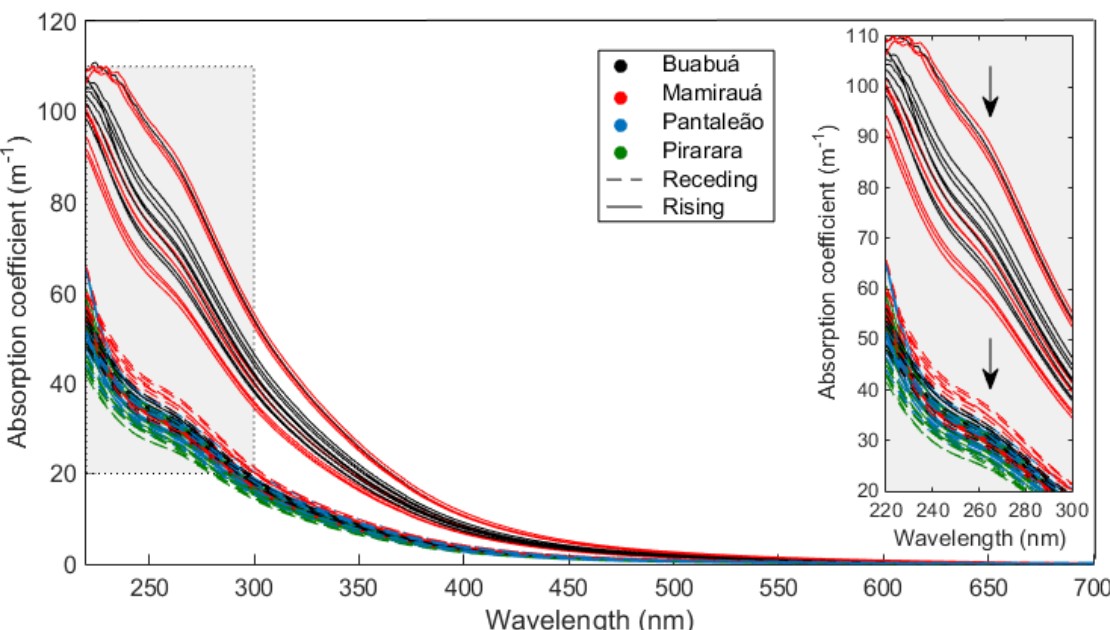

**Figure 4 - a**CDOM **absorption spectra collected in Buabuá (black), Mamirauá (red), Pantaleão (blue) and Pirarara (green) lakes during rising (solid line) and receding (dash line) water phase. The black arrows indicate the shoulder between 245 and 290 nm.**


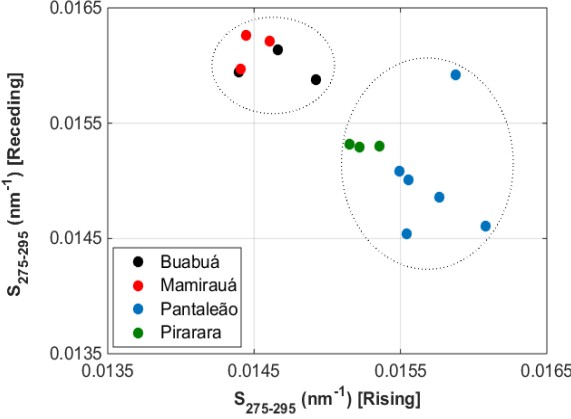

**Figure 5 – Dispersion diagram of average S₂₇₅₋₂₉₅ (nm⁻¹) at each hydrograph phase (rising and receding) and in all lakes. The dotted ellipsis represents the two groups identified.**



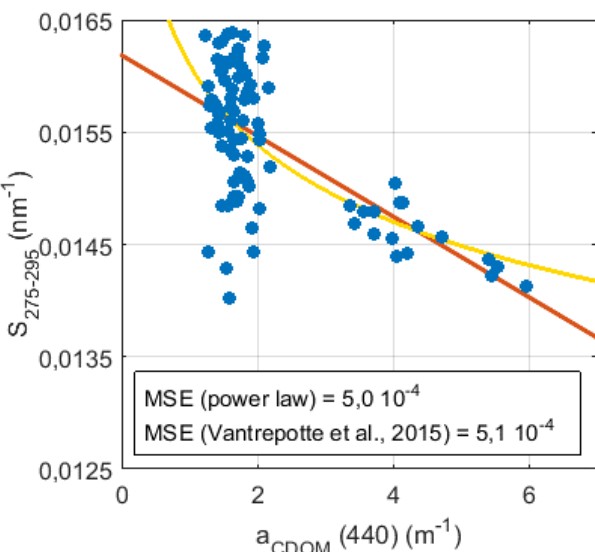

**Figure 6 - Adjustment of the model proposed by Vantrepotte et al. (2005) (in red) and adjustment of the proposed power-law model (in yellow).**

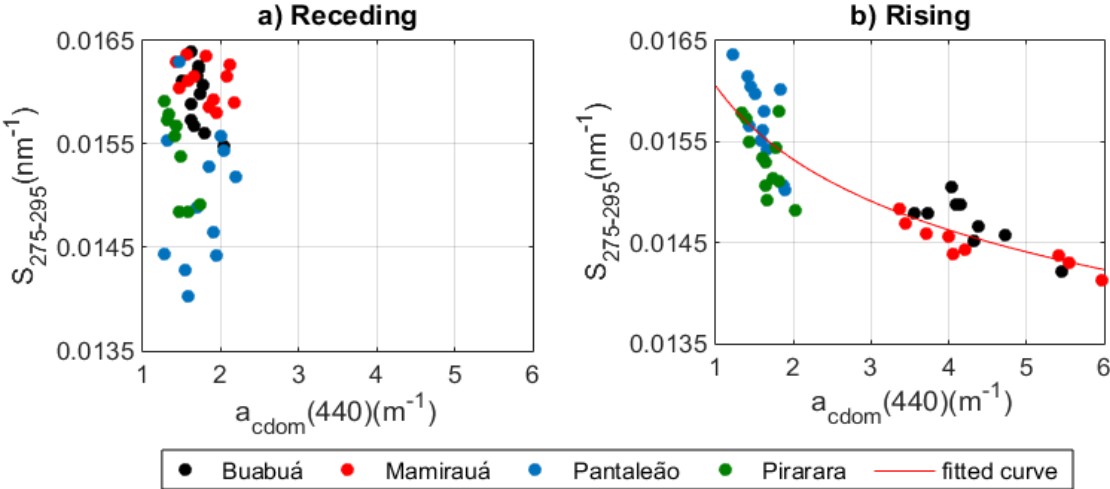

**Figure 7 - Scatterplot of $a_{CDOM}$ (440) (m$^{-1}$) versus $S_{275-295}$ (nm$^{-1}$) for a) receding and b) rising phases of the hydrograph.**





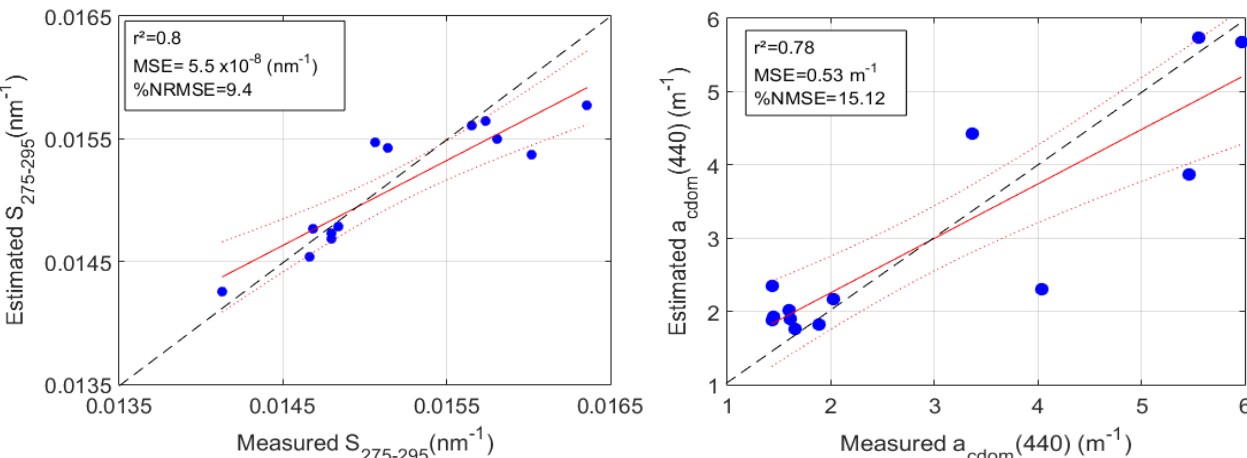


**Figure 8 – Measured versus estimated scatterplot of (left) S$_{275-295}$ (nm-1) and (right) a$_{CDOM}$ (440) (m$^{-1}$). Equation 8 was used to estimate S$_{275-295}$ (Y axis). Equation 9 was used to estimate a$_{CDOM}$(440) (Y axis). The red solid line indicates the regression line between measured and estimated values; the red double dotted lines are the 95% confidence interval; and the black dashed line is the 1:1 line.**




**Table 1 – Overview of the sampling points.**

| Sampling point | DOC [mgL$^{-1}$] | | | |
|---|---|---|---|---|
| | Rising water | | Receding water | |
| | March | April | July | August |
| Buabuá_1 | 9.0 | 7.2 | 4.0 | 4.7 |
| Buabuá_2 | 10.1 | 8.3 | 3.9 | 4.0 |
| Buabuá_3 | 7.6 | 7.7 | 4.0 | 4.2 |
| Buabuá_4 | | 8.2 | 3.8 | 4.1 |
| Buabuá_5 | | 7.7 | 4.1 | 4.0 |
| Buabuá_6 | | 8.1 | 3.9 | 3.9 |
| Mamirauá_1 | 9.6 | 9.4 | 4.5 | 4.2 |
| Mamirauá_2 | 7.4 | 9.5 | 4.5 | 3.7 |
| Mamirauá_3 | 7.3 | 9.7 | 4.9 | 3.7 |
| Mamirauá_4 | 8.0 | | 4.3 | 4.2 |
| Mamirauá_5 | 8.7 | | 4.3 | 4.0 |
| Mamirauá_6 | 7.7 | | 4.5 | 4.0 |
| Pantaleão_1 | 4.0 | 3.7 | 3.9 | 3.5 |
| Pantaleão_2 | 3.8 | 4.4 | 3.8 | 3.3 |
| Pantaleão_3 | 4.2 | 3.5 | 3.7 | 3.7 |
| Pantaleão_4 | 4.0 | 3.6 | 4.1 | 4.0 |
| Pantaleão_5 | 4.5 | 3.4 | 3.1 | 3.9 |
| Pantaleão_6 | 4.8 | 3.5 | 3.1 | 4.0 |
| Pirarara_1 | 3.8 | 3.7 | 3.9 | 4.7 |
| Pirarara_2 | 8.7 | 3.4 | 3.5 | 3.9 |
| Pirarara_3 | 3.7 | 3.4 | 3.9 | 4.0 |
| Pirarara_4 | 3.7 | 3.7 | | 3.9 |
| Pirarara_5 | 3.7 | 3.4 | | 3.9 |
| Pirarara_6 | 5.2 | 3.5 | | 4.8 |
