# Peer review of "Use of absorption optical indices to assess seasonal variability of dissolved organic matter in amazon floodplain lakes"

_Biogeosciences, 2019_

## Referee Comment (RC1) · Anonymous Referee #1 · 16 Oct 2019

Major comments

The authors measured CDOM parameters, i.e., aCDOM(440) and S275-295, of water samples collected from four lakes located at the Mamirauá Sustainable Development, Brazil. The lakes have different geographical settings: two of them are isolated perennial lakes surrounded by flood forests, while the others are lakes connected to the Japurá river. The authors found that levels of aCDOM(440) and values of S275-295 were different between rising and receding periods for the former lakes but not for the latter lakes. The authors found a power trend between aCDOM(440) and S275-295 for all lakes during the receding periods and concluded that S275-295 can be estimated

from aCDOM(440) during the receding periods. Additionally, the authors established a model to estimate aCDOM(440) from Rrs determined in stiu by optical sensors. From these results, the authors concluded that "The empirical model relating Rrs and aCDOM (440); aCDOM(440) and S275-295 provided robust statistics indicating the high potential of MSI sensor for estimating S275-295 during the rising water."

I think the measurements and data analyses in the manuscript were mostly reasonable. However, I could not understand why estimation of S275-295 from Rrs through the relationships with aCDOM(440) was necessary, because the motivation regarding with estimation of S275-295 from Rrs was not described/discussed. The authors referred papers by Fichot et al. (2003) and Vantrepott et al. (2015). Fichot et al. used S275-295 for a tracer of terrestrial DOM in the Arctic Ocean. Vantrepott et al. used S275-295 as a proxy of ratio of aCDOM to DOC. These previous studies, therefore, clearly mentioned the necessity to estimate the S275-295, in addition to and/or instead of aCDOM, from Rrs. However, in the manuscript, it seemed that the authors estimated S275-295 without clear purpose/motivation. The relationship between S275-295 and aCDOM(440) indicates that possible interpretation about environmental dynamics of CDOM by S275-295 estimated from aCDOM(440) and Rrs are the same with that by aCDOM(440) estimated from Rrs. In other words, the second main objectives of this study "compute S275-295 to examine its potential for distinguishing differences in DOM by comparing them in those two hydrograph phases" can be achieved only from aCDOM(440) without estimation of S275-295 from aCDOM(440). Thus, I think the estimation of S275-295 from aCDOM(440) is not necessary for this study. I think the comparison of aCDOM(440) estimated from Sentinel/MSI imagery and those from in situ measurements, rather than estimation of S275-295 from aCDOM(440), is much important and informative, even though the discussion about the observed relationship between S275-295 and aCDOM(440) is necessary.

Minor comments

Line 15: Please do not use abbreviation (MSI) for the first use.

Lines 46-48: I could not understand how Vantrepott et al. (2015) circumvent assumptions in Fichot et al. (2013), namely CDOM optical dominance in water and co-variation between CDOM and other particulate matter. Please explain the circumventor in detail with more logical manner.

Line 84: Please add more explanations about the methods by Mobley (1999) and Jorge et al. (2017b) for readers' convenience.

Line 93: How long the authors kept samples in the refrigerator?

Line 98: Lambda(ref) and Lambda(0) are usually the same in the equation described in spectral slope parameter (e.g., Bricaud et al., 1981).

Line 122: Line 84: Please add more explanations about the methods by Vantrepott et al. (2015) for readers' convenience.

Line 145: "cw" should be defined before use of the abbreviation.

Figure 2: It seemed that Figure 2 was not appeared (referred) in the text.

Figure 4: In addition to present Figure 4, addition of a figure having log scale of absorption coefficients on Y-axis may help readers' understanding.

Figure 5: I could not understand how the authors averaged the data. Please explain it.

Line 184: I could not understand the meaning of "high relationship". Please rephrase it.

Lines 213-220: I basically agree with the authors' discussion about differences in behaviors of CDOM parameters with rising/receding of the water between two types of the lakes. However, it may be possible to explain that the deviated behaviors observed in Buabuá and Mamirauá during the receding periods were simply due to the contribution of water from the Solimões River in which CDOM characteristics are largely different from the Japurá River and around the study region. Figure 1 clearly showed that colors, possibly affected by CDOM and particles, were largely different between

the Solimões River and the Japurá River. Thus, I think it's better to explain/discuss possible differences in CDOM parameters between two rivers and possible effects by the rivers, in particular the Solimões River, to CDOM parameters in the lakes during the rising/receding periods. In addition, it' s better to discuss why CDOM parameters in the lakes affected by the Japurá River were not changed during the rising period.

---

## Referee Comment (RC2) · Anonymous Referee #2 · 30 Nov 2019

Overall comments: The authors present a study of Amazon floodplain lakes during the rising and falling limbs of the hydrograph, using absorption spectra and simulated satellite remote sensing data to investigate the utility of remote sensing applications to the region. Overall, it's a very interesting dataset, with contrasting lakes (in terms of CDOM and hydrology). Remote sensing of inland waters is a rapidly growing field, and this is a valuable contribution, particularly calling out the need to explicitly address how to formulate models for complex waters with high CDOM and NAP.

I have a couple broad questions related to the premise of this study – if you are trying to test potential remote sensing algorithms, using simulated MSI or Landsat data, how

often are high quality images available that this could potentially be applied to? Clouds are a problem everywhere for these applications, but my impression is that they are an even more important factor in the Amazon and similar tropical regions. What is the satellite record like, particularly during the rising and falling limbs of the hydrograph when we would expect things to be most dynamic? Even in the absence of thick clouds, the high humidity, haze/thin clouds, and even smoke can be major barriers to reliable atmospheric corrections – since remote sensing of CDOM is so sensitive to atmospheric corrections, how do you think this would influence the usefulness of satellite applications for the region?

Second, I'm not sure why it was needed to use S275-295 instead of a440, in these circumstances. The spectral slope has been used mostly in coastal ocean studies with higher spectral resolution sensors, in cases where there were specific questions about the source of DOM (usually terrestrial versus marine). The goal of this study seems to be to trace bulk DOM, largely – a440 or other specific wavelengths have been used extensively for that type of application, in freshwaters. I question whether its appropriate to use the spectral slope for this type of question, environmental system, and sensor type, at least without further justification. Field measurements of spectral slope do provide additional information, but if its simply being estimated from a440 without additional parameters, then I don't think you can make more conclusions than you could from just a440.

Section 3.3: I honestly found this entire section fairly confusing. There needs to be more detail in the statistical description of models, to start. For instance, saying "validation results were satisfactory" is not sufficient. Also, it might be better to separate out the remote sensing model from the results on the relationship between spectral slope and a440, both in the text and in the figures. The questions being answered are completely separate: can field-measured a440 be used to predict S275-295 is a very different question than whether simulated remote sensing data can be used to predict a440. Also, if the ultimate goal is to derive s275-295 from remote sensing data, than

that needs to be presented, and for the propagation of error to be quantified somehow. Finally, it seems like the remote sensing model is only for the rising limb, not the receding – is that right? Or at least that it excludes the two lakes during the receding limb? If so, I think you need to further justify that decision – I understand that you cannot estimate spectral slope as easily, but there's not a clear reason why you can't estimate a440, which is still a very useful parameter.

There are a few issues with grammar and clarity, throughout the manuscript. While this was not so much that I had trouble understanding, the authors might consider an additional round of copy-editing. Overuse of commas, for instance, is sometimes an issue. I've pointed out some of these cases in specific comments, but not all.

Specific comments:

Intro: Line 26: there's more recent papers on the size of the DOM carbon budget that might be more appropriate – the whole special issue of Limnology and Oceanography Letters on carbon cycling of inland waters would be a good resource Line 27: Hastie et al 2019 on the Amazon carbon budget that incorporates aquatic cycling would be good to cite here and elsewhere Line 33: In what cases? Line 36: "being a relevant indicator. . ." that clause is probably not necessary – implicit in the rest of the sentence that it's a useful tool. Line 37: define CDOM. Also, CDOM is a concept of a pool of organic matter – the portion that absorbs light. It encompasses most of the various optical absorbance parameters, but it is usually better to specify what exact proxy is meant by CDOM. So, if you're referring to a specific proxy – Sr or a440 – its usually better to use that term than the broader "CDOM" category. Also, the sentence about Helms et al 2008 is a bit out of place – move to methods? Or wrap in a broader discussion of what the different CDOM/absorbance parameters mean and can tell us about the environment. Line 47: describe what a412 is, and aCDOM more generally. Line 50: Relatively few studies have looked at spectral slope remote sensing of inland waters, but there are MANY out there that look at a440, a412, a350, others. It'd be worthwhile to explain that this is commonly used. Mostly for Landsat, but there's a

few out there using Sentinel, and laying out for a non-remote sensing audience, as might be expected for Biogeosciences, that this is an established field would be useful. Line 62: I had a little trouble following this sentence. Consider revising. Line 64: commas are overused in this sentence Figure 1: Could you locate on the map where the discharge gauges are on each river? Also, since you don't refer to the roman numeral elsewhere in figures/tables, I would just put the name of each lake next to it, instead of the numerals Line 68: Briefly describe these criteria 69: This is the first time you've mentioned Landsat. If you're also including Landsat in your criteria and research, it should be discussed some in the intro, particularly since there's quite a bit of literature out there on remote sensing of CDOM using Landsat Line 73: So, these are connected all year long? There's not a time period when they aren't connected? At their lowest, do they consistently meet the criteria you mention above? Line 78: It looks like there are 24 stations, total, but 87 samples. Not 87 stations. Section 2.3.1: I am admittedly not as familiar with in situ Rrs methods as the other tools used here, but is there any information on quality control/error for this data set? Line 94: How long were samples stored before analysis? Line 95: blank corrected? Is this a single or dual beam spec? Given that the a440 was fairly low in some of these samples, what was the limit of detection with a 1 cm cuvette? Line 111: absorption at 440 nm in not that high – in fact, its not uncommon for it to be near detection limits in low-cdom lakes, when using a 1 cm cuvette, on many specs. Line 114: Please also specify that this is a one-way ANOVA on ranks – more informative if someone happens to not know the name of the statistical test offhand Line 118: I had trouble understanding this sentence, and how spectral slope was treated in relation to the hydrography. Please revise. Section 2.3.2: Generally, just describing that you made plots is not necessary in a description of statistical analyses. However, I would like to know what programs (or packages, if using R or python or the like) were used. Line 127: I agree that Monte Carlo is likely a good way to get around the limited number of samples, but was there any bias in how the calibration/validation data were split? I could imagine that might also influence results, if certain sites or seasons were over or under-represented in either dataset.

Line 140: The work done in Arctic rivers and estuaries are sometimes impacted by high sediment – see Matsuoka et al (multiple years – especially look at Mackenzie River and beaufort sea papers), Herrault et al 2017 (although this is the Yenisey, which has low sediment), and Griffin et al 2018. Brezonik et al 2015 also includes data from the St Louis River Estuary, that has both high CDOM and high sediment. While these papers do not propose the same model formation as you do, I believe most of them also mention the issue of sediment, and some include band ratios that incorporate NIR bands. Olmanson et al 2016 also uses the NIR landsat band. I think the rationale you present here is sound, and I have no problem with your model formulation, but I don't think you can claim others have not tried to develop models for similarly high NAP and high CDOM environments. I would also say that these sites are not necessarily "high" CDOM environments – that is, first, a somewhat subjective term. In addition, in my experience, CDOM is visible to the naked eye around 3 m-1 a440 – and only a portion of your samples reach that threshold. That's not necessarily the only way to claim that something is "high" CDOM, but it's an easy rule of thumb. Indeed, you acknowledge at the end of the discussion that this study only includes a small range of CDOM. Line 153: Are these averages across all sites? Please specify Line 155: Looking at Figure 3, at least some of the more highly colored lakes in Mar/Apr look like there's spatial variability within the lake. What did you do to conclude there was not spatial variability within lakes? Line 156: This sentence and the next I had some trouble following. Line 160: What were the differences? Specify Figure 5: Was there a statistical test or grouping analysis done to draw these circles, or just by sight? Also, it looks like the Buabua and Mamiraua do separate from the other lakes, even during the receding phase (and you point this out in the text). That contradicts the claim in Line 170 – I realize that refers specifically to a254, but given that the figure 4 shows all spectra, it's a little confusing Line 180: Please describe Sr in the methods Figure 6: Vantrepotte et al (2015), not 2005, in the caption. Also, what adjustment is being referred to? Also, please provide more detailed description of the statistics and model formulation (coefficients, etc), perhaps in a table. Line 194: I found this sentence

confusing. Please rephrase. Line 195: What model was selected??? Which data were included? Is this the model that was developed using Monte Carlo? Section 3.3: See major comment Figure 8: The equations are mislabeled. Line 213: But the previous paragraph just stated that hydrography was a controlling factor of CDOM! Is the flood pulse not a controlling factor of water level? If not, that needs to be explained more fully. Line 229: There are more recent studies on the Amazon about the role of DOC and incland waters on carbon cycling – Hastie et al 2018 (or maybe 2019) models that for the entire amazon, for instance.

---

## Author Comment (AC1) · 21 Dec 2019

Dear Referee #1,

We are grateful for the detailed and valuable comments on our manuscript. Special emphasis will be given on a proper discussion of S 275-295 biochemical significance and estimation. Please find our response to the comments in the link below, with the reviewer's comments in black and followed by our response in red.

On behalf of all co-authors,

Maria Paula da Silva

Please also note the supplement to this comment:
https://www.biogeosciences-discuss.net/bg-2019-324/bg-2019-324-AC1-
supplement.pdf

**Supplement:**

Major comments

The authors measured CDOM parameters, i.e., aCDOM(440) and S275-295, of water samples collected from four lakes located at the Mamirauá Sustainable Development, Brazil. The lakes have different geographical settings: two of them are isolated perennial lakes surrounded by flood forests, while the others are lakes connected to the Japurá river. The authors found that levels of aCDOM(440) and values of S275-295 were different between rising and receding periods for the former lakes but not for the latter lakes. The authors found a power trend between aCDOM(440) and S275-295 for all lakes during the receding periods and concluded that S275-295 can be estimated from aCDOM(440) during the receding periods. Additionally, the authors established a model to estimate aCDOM(440) from Rrs determined in stiu by optical sensors. From these results, the authors concluded that "The empirical model relating Rrs and aCDOM (440); aCDOM(440) and S275-295 provided robust statistics indicating the high potential of MSI sensor for estimating S275-295 during the rising water."

I think the measurements and data analyses in the manuscript were mostly reasonable. However, I could not understand why estimation of S275-295 from Rrs through the relationships with aCDOM(440) was necessary, because the motivation regarding with estimation of S275-295 from Rrs was not described/discussed. The authors referred papers by Fichot et al. (2003) and Vantrepott et al. (2015). Fichot et al. used S275295 for a tracer of terrestrial DOM in the Arctic Ocean. Vantrepott et al. used S275295 as a proxy of ratio of aCDOM to DOC. These previous studies, therefore, clearly mentioned the necessity to estimate the S275-295, in addition to and/or instead of aCDOM, from Rrs. However, in the manuscript, it seemed that the authors estimated S275-295 without clear purpose/motivation. The relationship between S275-295 and aCDOM(440) indicates that possible interpretation about environmental dynamics of CDOM by S275-295 estimated from aCDOM(440) and Rrs are the same with that by aCDOM(440) estimated from Rrs. In other words, the second main objectives of this study "compute S275-295 to examine its potential for distinguishing differences in DOM by comparing them in those two hydrograph phases" can be achieved only from aCDOM(440) without estimation of S275-295 from aCDOM(440). Thus, I think the estimation of S275-295 from aCDOM(440) is not necessary for this study. I think the comparison of aCDOM(440) estimated from Sentinel/MSI imagery and those from in situ measurements, rather than estimation of S275-295 from aCDOM(440), is much important and informative, even though the discussion about the observed relationship between S275-295 and aCDOM(440) is necessary.

Referee#1 Comment 1

The motivation in study $S_{275-295}$ is described in the manuscript (lines 31-37) and is based on the extra information that this parameter gives about DOM. $S_{275-295}$ has being used to estimate DOM molecular weight and sources (Helms et al., 2008; Spencer et al., 2008). Here it was used to understand the DOM dynamic in the four lakes of the floodplain during rising and receding water (lines 242-249).

Regarding the models proposed, we chose to use $a_{CDOM}$ as a proxy for $S_{275-295}$ once $a_{CDOM}$ is a parameter that is usually estimated when studying optical properties of water and, the main reason, because Vantrepotte et al. (2015) proved that their proposed method to estimate $S_{275-295}$ using $a_{CDOM}$ as proxy works better for water with different optical quality (e.g. influence of other optical active components greater than CDOM response) better than directly estimation of it from Rrs. More details about

Vantrepotte et al. (2015) method will be given in the method section as described in the reply to the comment 3. Also, testing the estimation of $S_{275-295}$ from Rrs in our dataset, the validation results are not good in our study area as well. The variation in $S_{275-295}$ does not result in variation in the Rrs.

[Figure]

Unfortunately, no images for the dates that we've the field data are available without cloud cover. Thus, we didn't apply the model to a Sentinel-2 image because the validation of estimated and measured $a_{CDOM}$ wouldn't be possible.

Minor comments Line 15: Please do not use abbreviation (MSI) for the first use.

Referee#1 Comment 2

Change will be incorporated as proposed:" The Rrs was applied to simulate visible bands of Multi-Spectral Instrument (MSI) and used in the proposed models."

Lines 46-48: I could not understand how Vantrepotte et al. (2015) circumvent assumptions in Fichot e tal. (2013), namely CDOM optical dominance in water and co-variation between CDOM and other particulate matter. Please explain the circumventor in detail with more logical manner.

Referee#1 Comment 3

According to Vantrepotte et al. (2015) the reflectance in visible bands is not fully related to changes in CDOM spectral slope in UV domain. These authors also found that the relationship between CDOM and $S_{275-295}$ is strong dependent of water optical quality being particularly conditioned by the dominant contribution of CDOM to the "water absorption budget as well as to a strong co-variation between dissolved and particulate matter dynamics". This conclusion was made based on the application of the model proposed by Fichot et al. (2012) in their broader data set, including different areas with different CDOM quality. On the other hand, their new model could estimate $S_{275-295}$ with better accuracy even when applied to another data set (not used in the parametrization of the model).

The sentence will be changed in the manuscript to: "…marine reflectance. However, the reflectance of water in the visible bands may not reflect changes in the spectral slope of CDOM in the UV domain as shown by Vantrepotte et al. (2015) applying the model in three coastal water regions. Then, these

authors proposed the use of $a_{CDOM}$ as a proxy for $S_{275-295}$ as it proved to be less affected by water optical quality and atmospheric correction."

Line84: Please add more explanations about the methods by Mobley (1999) and Jorge et al. (2017b) for readers' convenience.

**Referee#1 Comment 4**

The radiometric measurements to derive remote sensing reflectance were carried out for all sampling points, using three intercalibrated RAMSES–Trios sensors. The sensors measured above water radiance ($L_w$, $W \cdot m^{-2} \cdot sr^{-1} \cdot nm^{-1}$), sky radiance ($L_{SKY}$, $W \cdot m^{-2} \cdot sr^{-1} \cdot nm^{-1}$), and water surface irradiance ($E_s$, $W \cdot m^{-2} \cdot sr^{-1}$), between 350 and 900 nm. During the measurements, the sensors were positioned with azimuth angles between $90^0$ and $135^0$ in relation to the sun and a Zenith angle of $45^0$ to avoid sun glint effects (Mueller and Fargion, 2002). The measurement framework followed Mobley (1999). All of the measurements were made between 10:00 and 13:00 and at least 15 samples were obtained for each measured depth. The dataset was processed using MSDA_XE (TRIOS, 2018) and Matlab. The $R_{rs}$ estimate followed Mobley (1999), with sun glint correction based on each sampling point.

Line 93: How long the authors kept samples in the refrigerator?

Each field mission lasted around 12 days, with 8 days of sampling and the remaining days in transit. Considering that all samples were processed up to 4 days after returning to the lab, the samples were kept in the refrigerator for up to 14 days (8 samplings days + 2 days in transit + up to 4 days to be processed). During this time, water samples for CDOM absorption determination were kept in polypropylene bottles wrapped with black tape.

**Referee#1 Comment 5**

Line 98: Lambda(ref) and Lambda(0) are usually the same in the equation described in spectral slope parameter (e.g., Bricaud et al., 1981).

**Referee#1 Comment 6**

We agree. We apologize for the mistake. This will be changed in the manuscript as follow:"

$$a_{cdom}(\lambda) = a_{cdom}(\lambda_{ref}) \cdot e^{-S(\lambda - \lambda_{ref})}, \qquad (3)$$

where S is the spectral slope parameter ($nm^{-1}$) between the wavelength interval of $\lambda - \lambda_{ref}$ and $\lambda_{ref}$ is a reference wavelength (nm)."

Line 122: Line 84: Please add more explanations about the methods by Vantrepott et al. (2015) for readers' convenience.

**Referee#1 Comment 7**

This will be changed in the manuscript: "The model proposed by Vantrepotte et al. (2015) based on the ratio of $a_{CDOM}$ (412) and parameterized according to three coastal zones was tested to our data set using $a_{CDOM}$ (440), once Sentinel doesn't have a band in 412 nm. A simple power function (Equation 4) was also tested."

Line 145: "cw" should be defined before use of the abbreviation.

Referee#1 Comment 8

We agree. We apologize for the mistake. This will be changed in the manuscript as follow: "Thus, to determine $a_{CDOM}$ (440), the exponential of the ratio between bands 6 ($\lambda$central wavelength ($\lambda$cw) =740 nm) and 5 ($\lambda$cw =705 nm)…"

Figure 2: It seemed that Figure 2 was not appeared (referred) in the text.

Referee#1 Comment 9

We apologize for the mistake. The figure will be addressed in the section "2.3.2 Model calibration and validation".

Figure 4: In addition to present Figure 4, addition of a figure having log scale of absorption coefficients on Y-axis may help readers' understanding.

Referee#1 Comment 10

The Figure will be change in the manuscript.

Figure 5: I could not understand how the authors averaged the data. Please explain it.

Referee#1 Comment 11

Samples were collected during four field campaigns (March, April, July and August – Table 1) and two hydrograph phases. Figure 5 shows a comparison between $S_{275-295}$ during the rising and receding hydrograph phases in the four lakes. To be able to compare these two phases, we average $S_{275-295}$ (from the same sampling point) in March and April to plot the $S_{275-295}$ for the rising period and $S_{275-295}$ from July and August (from the same sampling point) to plot $S_{275-295}$ for the receding period. As example, $S_{275-295}$ for Buabua_1 was 0.0145 $nm^{-1}$ in March and 0.0148 $nm^{-1}$ in April, so for the rising period $S_{275-295}$ for Buabua_1 was computed as 0.01465 $nm^{-1}$. The same method was applied to compute $S_{275-295}$ during receding water: Buabua_1 was 0.0159 $nm^{-1}$ in July and 0.0164 $nm^{-1}$ in August, the resulting $S_{275-295}$ is 0.01615 $nm^{-1}$.

Line 184: I could not understand the meaning of "high relationship". Please rephrase it.

Referee#1 Comment 12

The sentence will be rewritten. In the manuscript it will addressed as "significant relationship".

Lines 213-220: I basically agree with the authors' discussion about differences in behaviors of CDOM parameters with rising/receding of the water between two types of the lakes. However, it may be possible to explain that the deviated behaviors observed in Buabuá and Mamirauá during the receding periods were simply due to the contribution of water from the Solimões River in which CDOM characteristics are largely different from the Japurá River and around the study region. Figure 1 clearly showed that colors, possibly affected by CDOM and particles, were largely different between the Solimões River and the Japurá River. Thus, I think it's better to explain/discuss possible differences in

CDOM parameters between two rivers and possible effects by the rivers, in particular the Solimões River, to CDOM parameters in the lakes during the rising/receding periods. In addition, it's better to discuss why CDOM parameters in the lakes affected by the Japurá River were not changed during the rising period.

Referee#1 Comment 13

We agree that the two main rivers (Solimoes and Japura) play an important role in the behavior of CDOM in the lakes and this was discussed in the manuscript (lines 215-241). However, besides the differences in water quality between rivers, we cannot exclude the impacts of water paths through the lakes, once the rivers' water overbank flooding though the forest during the rising phase of the hydrograph. An indicative of the forest as an important source of DOM is the DOC concentration. DOC concentration of Solimoes river is about 5.8 mg/L (Morreira-Turq et al., 2003), while DOC in the lakes is around 9 mg/L. Also, $S_{275-295}$ indicates high-molecular weight DOM present during rising period at Buabua and Mamiraua lakes, while DOM in rivers is expected to have low-molecular weight (Massicotte et al., 2017), indicating that additional DOM is being carry out to the lakes.

References

Fichot, C. G. and Benner, R. The spectral slope coefficient of chromophoric dissolved organic matter ($S_{275-295}$) as a tracer of terrigenous dissolved organic carbon in river-influenced ocean margins, Limnology and Oceanography, 57(5), 1453–1466, 2012.

Helms, J. R., Stubbins, A., Ritchie, J. D., Minor, E. C., Kieber, D. J., Mopper, K. Absorption spectral slopes and slope ratios as indicators of molecular weight, source, and photobleaching of chromophoric dissolved organic matter. Limonology and Oceanography, 53 (3), 955–969. https://doi.org/10.4319/lo.2008.53.3.0955, 2008

Massicotte, P., Asmala, E., Stedmon, C., Markager, S. Global distribution of dissolved organic matter along the aquatic continuum: Across rivers, lakes and oceans. Science of the Total Environment, 609, 180-191, ISSN 0048-9697, 2017.

Mobley, C.D. Estimation of the remote-sensing reflectance from above-surface measurements. Appl. Opt., 38, 7442–7455, 1999.

Moreira - Turcq, P., Seyler, P., Guyot, JL e Etcheber, H. Exportação de carbono orgânico do rio Amazonas e seus principais afluentes. Hydrol. Process., 17, 1329-1344, 2003

Mueller, J.L.; Fargion, G.S. Ocean Optics Protocols for Satellite Ocean Color Sensor Validation; Revision 3; NASA TM 2002-210004; NASA Goddard Space Flight Center: Greenbelt, MD, USA, 2002; p. 308.

Spencer, R. G., Aiken, G. R., Wickland, K. P., Striegl, R. G., Hernes, P. J. Seasonal and spatial variability in dissolved organic matter quantity and composition from the Yukon River basin, Alaska. Global Biogeochemical Cycles, 22 (4), 2008.

TRIOS. Trios sensor. 2018. Disponível em: https://www.trios.de/en/.

Vantrepotte, V., Danhiez, F. P., Loisel, H., Ouillon, S., Mériaux, X., Cauvin, A., Dessailly, D. CDOM-DOC relationship in contrasted coastal waters: implication for doc retrieval from ocean color remote sensing observation. Optics Express, 23 (1), 33. https://doi.org/10.1364/oe.23.000033, 2015.

---

## Author Comment (AC2) · 21 Dec 2019

Dear Reviewer #2,

We appreciate your comments and your valuable revision. Responding to your comments, the text has been revised. Please find in the link below our detailed response to your comments and suggestions. Your comments are in black followed by our reply in red.

On behalf of all co-authors,

Maria Paula da Silva

[Figure]

Please also note the supplement to this comment:
https://www.biogeosciences-discuss.net/bg-2019-324/bg-2019-324-AC2-
supplement.pdf

**Supplement:**

I have a couple broad questions related to the premise of this study – if you are trying to test potential remote sensing algorithms, using simulated MSI or Landsat data, how often are high quality images available that this could potentially be applied to? Clouds are a problem everywhere for these applications, but my impression is that they are an even more important factor in the Amazon and similar tropical regions. What is the satellite record like, particularly during the rising and falling limbs of the hydrograph when we would expect things to be most dynamic? Even in the absence of thick clouds, the high humidity, haze/thin clouds, and even smoke can be major barriers to reliable atmospheric corrections – since remote sensing of CDOM is so sensitive to atmospheric corrections, how do you think this would influence the usefulness of satellite applications for the region?

Referee#2 Comment 1

By the time we started our study, there was only one satellite in the Sentinel constellation, therefore the probability of having cloud-free images was not high during the rainy season. Nowadays, however, there are two satellites collecting data every 5 days, which increases the probability of, at least one partially cloud-free image. The present satellite missions are constellation oriented and there has been a huge effort in the signal processing algorithms oriented towards multisensor cross-calibration and harmonization (Claverie et al., 2018). Moreover, sensors have improved their sensitivity to the low signal having improved signal-to-noise ratios and higher radiometric resolution. Moreover, the best window for acquiring satellite images in the Amazon Basin (Martins et al., 2018) covers most of the period encompassing the rising and falling limbs of the Amazon River flood pulse.

We agree that using satellite remote sensing for monitoring CDOM is really challenging not only due to the low signal under a huge atmosphere interference but also due to glint effects and diffuse scatter of the surrounding forest. In the attempt of overcoming those limitations, we can push the knowledge ahead. The present study is just a little brick in this attempt.

Despite all the hard work in a very defying environment, having done numerous missions to get such a small amount of data, we think that without including remote sensing tools in the study of dissolved organic matter pathways from the forest to the water we will not fully understand the diversity and processes linking this complex environment.

In this first brick, we think we have proven that it is possible to model the relationship between UV slope and $a_{CDOM}$ having a sensor with the spectral features of Sentinel/MSI. We recognize the weaknesses of the model mainly due to the sample size. But it is a first brick in the wall. This brick can be even replaced for a harder brick, but it is a brick.

Second, I'm not sure why it was needed to use S275-295 instead of a440, in these circumstances. The spectral slope has been used mostly in coastal ocean studies with higher spectral resolution sensors, in cases where there were specific questions about the source of DOM (usually terrestrial versus marine). The goal of this study seems to be to trace bulk DOM, largely – a440 or other specific wavelengths have been used extensively for that type of application, in freshwaters. I question whether its appropriate to use the spectral slope for this type of question, environmental system, and sensor type, at least without further justification. Field measurements of spectral slope do provide additional information, but if its simply being estimated from a440 without additional parameters, then I don't think you can make more conclusions than you could from just a440.

Referee#2 Comment 2

As there is an increased interest in assessing DOM quality and considering that literature have reported that there is a relationship between S value in the UV region and DOM properties (Helms et al., 2008; Peacock et al., 2014) (not only to its origin but also degradation state, for instance), we thought that if we could use optical data (e.g. absorption) derived from satellite, one could register changes in DOM quality patterns along time in response to the flood pulse. We have seen more and more studies focusing on the study of DOM quality using absorption data, especially using field Spectrolyzers. In our paper, we have tried to push the remote sensing data in this direction, taking into account the diversity and dimension of the Amazon basin, where it is hard to scaling up in situ data to broader regions. We know that the use of satellite information in the analysis of DOM quality is even more challenging once you have the influence of other parameters as the atmosphere, sensor and targets around. However, some studies (Vantrepotte et al., 2015; Fichot et al, 2013) have already shown that it is possible to estimate the spectral slope. Going in this direction, here we proposed a model to estimate $S_{275-295}$ from remote sensing data in a very complex system, regarding optical characteristics.

Regarding the different information that CDOM and S can tell, while $a_{CDOM}$ gives a quantitative parameter (the intensity of $a_{CDOM}$ signal is proportional to DOC concentration), S is related to properties of the DOM (Helms et al., 2008).

Section 3.3: I honestly found this entire section fairly confusing. There needs to be more detail in the statistical description of models, to start. For instance, saying "validation results were satisfactory" is not sufficient.
Referee#2 Comment 3

Authors agree with reviewer and have included more statistical information regarding model validation. The sentence will be changed to: "Validation results showed a good explanation of the model´s variance ($r^2$=0.8) and with values predicted by the model close to the observed values (%NRMSE=9.4), indicating the feasibility of estimating $S_{275-295}$ from $a_{CDOM}$ (440) (Figure 8a)."

Also, it might be better to separate out the remote sensing model from the results on the relationship between spectral slope and a440, both in the text and in the figures.
The questions being answered are completely separate: can field-measured a440 be used to predict S275-295 is a very different question than whether simulated remote sensing data can be used to predict a440.
Referee#2 Comment 4

We agree that those are two separated questions. But our premise is that if we can use field-measured a440 to predict $S_{275-295}$, we would only need to estimate a440 from satellite Rrs and apply the model to every water pixel to spatially map S. That is why they are not separated.

Also, if the ultimate goal is to derive s275- 295 from remote sensing data, than that needs to be presented, and for the propagation of error to be quantified somehow.
Finally, it seems like the remote sensing model is only for the rising limb, not the receding – is that right? Or at least that it excludes the two lakes during the receding limb? If so, I think you need to further justify that decision – I understand that you cannot estimate spectral slope as easily, but there's not a clear reason why you can't estimate a440, which is still a very useful parameter.
Referee#2 Comment 5

Yes, our ultimate goal was to assess the potential of estimating S using remote sensing data. The results, however, indicate that the model only works for the rising limb, which

is not a bad thing, because the largest differences among $a_{CDOM}$ and S happen during the rising limb. During the receding limp and low water DOM pools become more homogeneous again due to both photodegradation and biodegradation as shown in Figure 3.

There are a few issues with grammar and clarity, throughout the manuscript. While this was not so much that I had trouble understanding, the authors might consider an additional round of copy-editing. Overuse of commas, for instance, is sometimes an issue. I've pointed out some of these cases in specific comments, but not all.
Referee#2 Comment 6

Authors apologize for the poor English version which was revised.

**Specific comments:**
Line 26: there's more recent papers on the size of the DOM carbon budget that might be more appropriate – the whole special issue of Limnology and Oceanography Letters on carbon cycling of inland waters would be a good resource
Referee#2 Comment 7

We would like to thank the reviewer for the suggestion. We include the following references in line 26: " DOM represents the largest pool of organic carbon in the aquatic environment (Cole et al., 2007; Seekell et al., 2018; Tranvik et al., 2009) and…"

And in line 232: "In the rising phase, the water coming mainly from the Solimões river undergoes overbank flooding (Figure 1c), overtopping its channel and flowing across the litter through the forest before reaching the lakes (Junk, 1989). The tree-DOM may be a n import source of organic matter carried to the lakes during this event (Van Stan and Stubbins, 2018) carrying a great amount of organic matter accumulated during the lower water season. "

Line 27: Hastie et al 2019 on the Amazon carbon budget that incorporates aquatic cycling would be good to cite here and elsewhere
Referee#2 Comment 8

Thank you for the suggestion. The study analyses the flux of C between river and floodplain in Amazon during wet and dry periods. The authors found that net ecosystem productivity (NEP) is higher in wet periods than in lowest discharge levels. Now it is incorporated in the text: "… playing a vital role in primary productivity of aquatic ecosystems and consequently fisheries and other food webs (Hastie et al., 2019; Maia and Volpato, 2013; Volpato et al., 2004)."

Line 33: In what cases?
Referee#2 Comment 9

Here we refer to cases in which only DOC concentration can limit the study of the seasonal variation in DOM composition and origin because it is only related to the bulk DOM. To clarify the sentence, it will be changed to: "However, simple measurement of DOC concentration can limit the study of the seasonal variation in the DOM composition and origin since it is related only to the bulk DOM (Jaffé et al., 2008). Quality parameters are needed to better understand DOM dynamics such as ultraviolet (UV) and visible absorption measurements, fluorescence, which are an alternative for high costly laboratory analysis (Li and Hur, 2017)."

Line 36: "being a relevant indicator: : :" that clause is probably not necessary – implicit in the rest of the sentence that it's a useful tool.
Referee#2 Comment 10

The sentence will be changed in the manuscript and it is now in the section 2.3.2 Spectral slope determination: "Helms et al., (2008) have shown that the spectral slope in the range of 275 and 295 nm ($S_{275-295)}$ is an indicator of DOM molecular weight and a tracer of degradation processes."

Line 37: define CDOM. Also, CDOM is a concept of a pool of organic matter – the portion that absorbs light. It encompasses most of the various optical absorbance parameters, but it is usually better to specify what exact proxy is meant by CDOM. So, if you're referring to a specific proxy – Sr or a440 – its usually better to use that term than the broader "CDOM" category.
Referee#2 Comment 11

The sentence will be changed to: "A study carried out on Mississippi and Atchafalaya River revealed that the rates of water exchange between river and floodplain increase DOC concentrations and alter DOM composition (Shen et al., 2012). The authors have reached to this conclusion based on measurements of the absorption coefficient of colored dissolved organic matter (CDOM) at 350 nm ($a_{CDOM}$ (350)), $a_{CDOM}$ spectral slope ($S_{275-295}$) and DOC concentration.  DOM optical properties (Spencer et al. 2008) such as $a_{CDOM}$ (350) presented higher values during the spring flush at Yukon River basin due to the remarkably high content of aromatic vascular plant material derived from surface soil and litter layers."

Also, the sentence about Helms et al 2008 is a bit out of place – move to methods? Or wrap in a broader discussion of what the different CDOM/absorbance parameters mean and can tell us about the environment.
Referee#2 Comment 12

The sentence was changed to the section "2.3.2 Spectral slope determination" as follow:" Helms et al., (2008) have shown that the spectral slope calculated in the range of 275 and 295 nm ($S_{275-295}$) is an indicator of DOM molecular weight and a tracer of degradation processes. In the present study, the spectral slope in the interval between 275 and 295 nm ($S_{275-295}$) was computed according to the Equation 3 (Bricaud et al., 1981), using non-linear fit (Helms et al., 2008). This function describes the $a_{CDOM}$ ($\lambda$) behavior along the electromagnetic spectrum and is expressed as:"

Line 47: describe what a412 is, and aCDOM more generally.
Referee#2 Comment 13

The meaning of $a_{CDOM}$ ($\lambda$) – absorption coefficient (a) of colored dissolved organic matter at the wavelength $\lambda$ are now described above (Please see comment 8).

Line 50: Relatively few studies have looked at spectral slope remote sensing of inland waters, but there are MANY out there that look at a440, a412, a350, others. It'd be worthwhile to explain that this is commonly used. Mostly for Landsat, but there's a few out there using Sentinel, and laying out for a non-remote sensing audience, as might be expected for Biogeosciences, that this is an established field would be useful.
Referee#2 Comment 14

The text will be changed to: "In order to study DOM dynamics in wide spatial-temporal scale satellite images have been assessed as a source of optical information about CDOM. Many studies (Kutser et al., 2005; Brezonik et al., 2015) have used Landsat images for studying aCDOM at the different wavelengths, more commonly at 440 and 420. In recent years the availability of Multi-Spectral Instrument (MSI) images, on board of the Sentinel-2A (June/2015) and Sentinel-2B (March/2017), has expanded the potential of remote sensing application for DOM monitoring. According to Toming et al. (2016) MSI-Sentinel suitability for studying DOM. However, only a few studies have looked at the spectral slope of DOM. In a Pan-Arctic study, Fichot et al. (2013) showed that $S_{275-295}$ can be directly estimated from satellite images using a multi-linear parameterization of MODIS marine reflectance. However, the relationship proposed by Fichot et al. (2013) between Remote Sensing Reflectance (Rrs) and $S_{275-295}$ is based on two assumptions: i) CDOM optical dominance in 45 water bodies; ii) co-variation between CDOM and other particulate matter. To circumvent these assumptions, Vantrepotte et al. (2015) investigating the relationship between CDOM and DOC showed that S275-295 can be estimated from MODIS data using aCDOM (412) as proxy. Nonetheless, both studies (Fichot et al., 2013; Vantrepotte et al., 2015) used MODIS data whose spatial resolution (250-1000 m) restricts the application to inland water studies."

Line 62: I had a little trouble following this sentence. Consider revising.
Referee#2 Comment 15

The sentence will be changed to: "The seasonal flood is caused by both the rainfalls (in upper Amazon basin and locally - from December to May, with an average of 300 mm/month) and by the annual melt of the Andean cordillera during the austral summer (Junk, 1989)."

Line 64: commas are overused in this sentence
Referee#2 Comment 16

The sentence will be changed to: "The yearly MSDR flood pulse causes (Queiroz, 2007), in average, 12 meters amplitude in the water level between the dry (September to November) and the flood season (May and mid-July)."

Figure 1: Could you locate on the map where the discharge gauges are on each river? Also, since you don't refer to the roman numeral elsewhere in figures/tables, I would just put the name of each lake next to it, instead of the numerals
Referee#2 Comment 17

The discharge gauges are outside the image showed in the manuscript. The Solimões´s station is upstream from the Solimões river (2.49° S, 66.06° W) and the Japurá station upstream from Japurá river (1.86° S, 65.60° W). Regarding the lakes´ number, we appreciate the suggestion but we think that the image stays cleaner showing the lakes´ name in the legend.

Line 68: Briefly describe these criteria 69: This is the first time you've mentioned Landsat. If you're also including Landsat in your criteria and research, it should be discussed some in the intro, particularly since there's quite a bit of literature out there on remote sensing of CDOM using Landsat.
Referee#2 Comment 18

The Landsat was included in the criteria because the study cited (Jorge et al., 2017a) analyzed Landsat data. Since here we only focused on Sentinel, we decided to remove

it. The sentence will be changed to:" The lakes were selected according to criteria defined in Jorge et al. (2017a) to guarantee access to them throughout the hydrological year and sizes compatible to the spatial resolution of satellite sensors (MSI/ Sentinel-2A)."

Line 73: So, these are connected all year long? There's not a time period when they aren't connected? At their lowest, do they consistently meet the criteria you mention above?
Referee#2 Comment 19

Mamirauá, Pirarara and Pantaeão are connected to the river during the whole year, even during the dry season. Exceptions may occur during extreme dry years, when the lakes´ level is extreme low. Buabuá tends to lose connection in the dry month (September).

O Pirará e o Pantaleão são conectados e tem acesso o ano todo. Em anos de secas extremas, eles podem perde conexão durante as cotas mínimas mas a norma é que haja conexão. O Mamirauá e o Buá-Bua tendem a perder conexão no mês de mínima.

Line 78: It looks like there are 24 stations, total, but 87 samples. Not 87 stations. Section 2.3.1:
I am admittedly not as familiar with in situ Rrs methods as the other tools used here, but is there any information on quality control/error for this data set?
Referee#2 Comment 20

You are right. The sentence will be changed to:" in total 87 samples were collected among the lakes and seasons"

Regarding quality control, all measurements were taken following the IOCCG protocols and previous experience on Amazon Floodplain lakes reported in Sander de Carvalho (2015). Unfortunately due to the lack of resources no replicas were taken for CDOM absorption measurements. For radiometric measurements, errors are within the 30 % expected for *in situ* remote sensing sampling.

Line 94: How long were samples stored before analysis?
Referee#2 Comment 21

Each field mission lasted around 12 days, with 8 days of sampling and the remaining days in transit. Considering that all samples were processed up to 4 days after returning to the lab, the samples were kept in the refrigerator for up to 14 days (8 samplings days + 2 days in transit + up to 4 days to be processed).

Line 95: blank corrected? Is this a single or dual beam spec? Given that the a440 was fairly low in some of these samples, what was the limit of detection with a 1 cm cuvette?
Referee#2 Comment 22

The samples were corrected by blank measurements and dual beam spec. was used.

Line 111: absorption at 440 nm in not that high – in fact, its not uncommon for it to be near detection limits in low-cdom lakes, when using a 1 cm cuvette, on many specs.
Referee#2 Comment 23

We chose this wavelength as a compromise between absorption signal from the spectrometer and wavelength available in Sentinel as one should expected better correlation between information in the same wavelength. Also, as describe, it is a common to choose this wavelength in remote sensing studies.

Line 114: Please also specify that this is a oneway ANOVA on ranks – more informative if someone happens to not know the name of the statistical test offhand.
Referee#2 Comment 24

The sentence will be changed to:" Kruskal Wallis test (oneway ANOVA on rank) with a significance level of 95% was applied to test the existence of significant…"

Line 118: I had trouble understanding this sentence, and how spectral slope was treated in relation to the hydrography. Please revise.
Referee#2 Comment 25

We changed the sentence in the manuscript to: "The mean $S_{275-295}$ among months of the same hydrograph phases (e.g. July and August for receding; March and April for rising) was computed for each sampling point to analyze their variability within each lake and phase. "

Section 2.3.2: Generally, just describing that you made plots is not necessary in a description of statistical analyses. However, I would like to know what programs (or packages, if using R or python or the like) were used.
Referee#2 Comment 26

We changed this part in the manuscript: "…phase. The statistical analysis were performed using the software Matlab (Mathworks, Natick, MA, USA)."

Line 127: I agree that Monte Carlo is likely a good way to get around the limited number of samples, but was there any bias in how the calibration/validation data were split? I could imagine that might also influence results, if certain sites or seasons were over or under-represented in either dataset.
Referee#2 Comment 27

We tried to select the most representative model out of 100,000 runs by applying the methodology to select the model described in Augusto-Silva et al. (2014). Note that we didn't select the model with best performance, but the one that was in the range of most representative mean square error and coefficients. Also, the Monte Carlo was only run in the model selection and not validation. In this way, we tried to minimize the bias.

Line 140: The work done in Arctic rivers and estuaries are sometimes impacted by high sediment – see Matsuoka et al (multiple years – especially look at Mackenzie River and beaufort sea papers), Herrault et al 2017 (although this is the Yenisey, which has low sediment), and Griffin et al 2018. Brezonik et al 2015 also includes data from the St Louis River Estuary, that has both high CDOM and high sediment. While these papers do not propose the same model formation as you do, I believe most of them also mention the issue of sediment, and some include band ratios that incorporate NIR bands. Olmanson et al 2016 also uses the NIR landsat band. I think the rationale you present here is sound, and I have no problem with your model formulation, but I don't think you can claim others have not tried to develop models for similarly high NAP and high CDOM environments. I would also say that these sites are not necessarily

"high" CDOM environments – that is, first, a somewhat subjective term. In addition, in my experience, CDOM is visible to the naked eye around 3 m-1 a440 – and only a portion of your samples reach that threshold. That's not necessarily the only way to claim that something is "high" CDOM, but it's an easy rule of thumb. Indeed, you acknowledge at the end of the discussion that this study only includes a small range of CDOM.
Referee#2 Comment 28

We appreciate the suggestion and we agree. The text will be change to:" Once the relationship between $S_{275-29}$ and $a_{CDOM}$ (440) was modeled, another algorithm was calibrated and validated to estimate $a_{CDOM}$ (440) using the simulated MSI Rrs. There is some effort in the literature to propose models to estimate $a_{CDOM}$ in complex environments, regarding high CDOM and Non-Algal Particle (NAP) contribution to the Rrs (Matsuoka et al., 2009; Matsuoka et al., 2012). In this study, we propose a new model for estimating $a_{CDOM}$ (440) introducing a ratio between near infrared bands to remove NAP contribution from its inorganic fraction. The rational for introducing this ratio is the null signal of CDOM and the dominance of NAP in this wavelength range (Kirk, 2011)"

Line 153: Are these averages across all sites? Please specify
Referee#2 Comment 29

Yes, the averages are across all sites. The text will be changed to:" The highest amplitude of $a_{CDOM}$ (440) in the entire data set (e.g. across all sites) occurred in March (1.22 to 5.46 $m^{-1}$) and April (1.60 to 5.97 $m^{-1}$), with averages of 2.56 and 3.01 $m^{-1}$, respectively"

Line 155: Looking at Figure 3, at least some of the more highly colored lakes in Mar/Apr look like there's spatial variability within the lake. What did you do to conclude there was not spatial variability within lakes?
Referee#2 Comment 30

We based our conclusion in the Kruskal Wallis results, that showed no difference between lakes when all the data set was analyzed.

Line 156: This sentence and the next I had some trouble following.
Referee#2 Comment 31

The sentences will be changed to: "The water level during the sampling campaign in the rising and receding phase was almost the same (mean:30.04 ± 1.38 m). However, at the rising phase, high variability (CV: 52.45%) of $a_{CDOM}$ (440) occurred, while in the receding $a_{CDOM}$ (440) variability (CV: 14.74%) was much smaller."

Line 160: What were the differences? Specify
Referee#2 Comment 32

The sentence was rewritten:" Kruskal Wallis results using samples from all lakes and dates indicated that there are significant differences (p<0.001) in aCDOM (440) between lakes and hydrograph phases. After the removal of Buabuá and Mamirauá samples acquired in March and April (rising), Kruskal Wallis results showed no significant differences in $a_{CDOM}$ (440) values (p=0.51). The two runs indicate that DOM

at Buabuá and Mamirauá, during the rising phase have a much higher absorption at 440 nm than those of the remaining lakes and months."

Figure 5: Was there a statistical test or grouping analysis done to draw these circles, or just by sight? Also,it looks like the Buabua and Mamiraua do separate from the other lakes, even during the receding phase (and you point this out in the text). That contradicts the claim in
Referee#2 Comment 33

The circles were draw by sight. This data is related to the mean $S_{275-295}$ between the two months of the same hydrograph phase (please see comment 21). Indeed $S_{275-295}$ varies between hydrograph phases, but this shifts are not follow by $a_{CDOM}$. In Figure 6 you can see how this two variables correlate.

Line 170 – I realize that refers specifically to a254, but given that the figure 4 shows all spectra, it's a little confusing
Referee#2 Comment 34

We opted to show the entire spectra to also discuss it shape. A254 was also chosen to explain the differences in the magnitude of the spectra because it is not close to the limits of instrument detection and it is a common parameter analyzed for aromaticity. Even though aromaticity was outside the scope of the study, one could use it to compare with other sites.

Line 180: Please describe Sr in the methods
Referee#2 Comment 35

It was described at "2.3.2 Spectral slope determination": "The spectral slope ratio (SR) between the wavelength intervals of 275-295 nm on 350-400 nm was also computed in the same away describe in Equation 3". And the sentence on line 180 was changed to:" $S_R$ was applied to trace DOM sources in a tropical river-wetland system (Dalmagro et al., 2017)."

Figure 6: Vantrepotte et al (2015), not 2005, in the caption. Also, what adjustment is being referred to? Also, please provide more detailed description of the statistics and model formulation (coefficients, etc), perhaps in a table.
Referee#2 Comment 36

It has been changed in the caption: "Figure 6 - Adjustment of the model proposed by Vantrepotte et al. (2015, Equation 7) (in red) and adjustment of the proposed power-law model described in Equation 6 (in yellow)." The coefficients of the equation were add on line 188 in the section "3.3 Seasonal relationship between $a_{CDOM}$ and $S_{275-295}$" as follow, since the information regarding model fitting is already expressed in the figure:" The model proposed by Vantrepotte et al. (2015) was tested using the entire data set (coefficients of equation 0.05, 0.10, 3.06 and 0.0), but a power-law function provided a better fit (Figure 6).

Line 194: I found this sentence confusing. Please rephrase.
Referee#2 Comment 37

The sentence was re written: "As in the receding phase $a_{CDOM}$ values are very similar between lakes and no correlation was found between $a_{CDOM}$ (440) and $S_{275-295}$ the model was developed just with the samples acquired in the rising phase.

Line 195: What model was selected??? Which data were included? Is this the model that was developed using Monte Carlo?
Referee#2 Comment 38

The model selected was to estimate $S_{275-295}$ from $a_{CDOM}(440)$. The model selection in the sentence states for the coefficients chosen based on the Monte Carlo simulation. For this model, only data from the rising phase were selected. The sentence has been changes in the manuscript: "The selected model to estimate $S_{275-295}$ from $a_{CDOM}$ (440), developed using Monte Carlo and data from rising phase, shows a satisfactory fit (MSE<0.0001) and is described in Equation 6:"

Section 3.3: See major comment
Referee#2 Comment 39

Please see reply to comment 3

Figure 8: The equations are mislabeled.
Referee#2 Comment 40

Now it has been changes: "Figure 8 – Measured versus estimated scatterplot of (left) $S_{275-295}$ ($nm^{-1}$) and (right) $a_{CDOM}$ (440) ($m^{-1}$). Equation 4 was used to estimate $S_{275-295}$ (Y axis). Equation 5 was used to estimate $a_{CDOM}$(440) (Y axis). The red solid line indicates the regression line between measured and estimated values; the red double dotted lines are the 95% confidence interval; and the black dashed line is the 1:1 line."

Line 213: But the previous paragraph just stated that hydrography was a controlling factor of CDOM! Is the flood pulse not a controlling factor of water level? If not, that needs to be explained more fully.
Referee#2 Comment 41

What we wanted to state here is that the water level is not the major factor controlling $a_{CDOM}$ since at the same water level we have different values of $a_{CDOM}$ (440). Thus the phase of hydrograph (related to the flood pulse) is a key point in the $a_{CDOM}$ dynamic. The sentence will be change to:" The water level in the floodplain is quite similar between the rising and receding seasons, suggesting that the flood pulse is the major factor explaining the variability of those optical variables"

Line 229: There are more recent studies on the Amazon about the role of DOC and inland waters on carbon cycling – Hastie et al 2018 (or maybe 2019) models that for the entire amazon, for instance.
Referee#2 Comment 42

We appreciate the suggestion, but we decide for another reference, more recent than before and carried out during seven years (1994–2000). "Considering that in the Amazon basin, DOC accounts for 70% of total organic matter and that floodplain areas are relevant sources of DOC to the Solimões/Amazon River (Morreira-Turq et al., 2003)"

References

Brezonik, P. L.; Olmanson, L. G.; Finlay, J. C.; Bauer, M. E. Factors affecting the measurement of CDOM by remote sensing of optically complex inland waters. Remote Sensing of Environment, v. 157, p. 199–215, 2015.

Claverie, M.; Ju, J.; Masek, J. G., Dungan, J. L.; Vermote, E. F.; Roger, J.; Skakun, S. V.; Justice, C.The Harmonized Landsat and Sentinel-2 surface reflectance data set, Remote Sensing of Environment, 219, 145-161, 2018.

Dalmagro, H. J., Zanella de Arruda, P.H., Vourlitis, G.L., Lathuillière, M.J., DE S. Nogueira, J., Couto, E.G., Johnson, M.S. Radiative forcing of methane fluxes o sets net carbon dioxide uptake for a tropical flooded forest. Global Change Biology, 4, 1967–1981, 2019.

Fichot, C. G., Kaiser, K., Hooker, S. B., Amon, R. M., Babin, M., Bélanger, S., Walker, S. A., Benner, R. Pan-Arctic distributions of continental runoff in the Arctic Ocean. Scientific reports, 3, 1053. https://doi.org/10.1038/srep01053, 2013.

Hastie, A, Lauerwald, R, Ciais, P, Regnier, P. Aquatic carbon fluxes dampen the overall variation of net ecosystem productivity in the Amazon basin: An analysis of the interannual variability in the boundless carbon cycle. Glob Change Biol. 25, 2094–2111, 2019.

Helms, J. R., Stubbins, A., Ritchie, J. D., Minor, E. C., Kieber, D. J., Mopper, K. Absorption spectral slopes and slope ratios as indicators of molecular weight, source, and photobleaching of chromophoric dissolved organic matter. Limonology and Oceanography, 53 (3), 955–969. https://doi.org/10.4319/lo.2008.53.3.0955, 2008.

Jaffé, R., McKnight, D., Maie, N., Cory, R., McDowell, W. H., Campbell, J. L. Spatial and temporal variations in DOM composition in ecosystems: The importance of long-term monitoring of optical properties. Journal of Geophysical Research: Biogeosciences, 113 (G4), 2008.

Jorge, D. S. F., Barbosa, C. C., Affonso, A. G., Novo, E. M. L. DE M. Spatial-temporal characterization of optical properties of 4 lakes in the Mamirauá Sustainable Development Reserve - AM (MSDR). In: Anais XVIII Simpósio Brasileiro de Sensoriamento Remoto, Santos - SP, 2017a.

Kutser, T.; Pierson, D. C.; Kallio, K. Y.; Reinart, A.; Sobek, S. Mapping lake CDOM by satellite remote sensing. Remote Sensing of Environment, v. 94, n. 4, p. 535-540, 2005.

Li, P., Hur, J. Utilization of UV-Vis spectroscopy and related data analyses for dissolved organic matter (DOM) studies: A review. Critical Reviews in Environmental Science and Technology, 47 (3), 131-154, 2017.

Maia, C. M., Volpato, G. L. Environmental light color affects the stress response of Nile tilapia. Zoology, 116 (1), 2013.

Martins, V. S., Novo, E. M., Lyapustin, A., Aragao, L. E., Freitas, S. R., & Barbosa, C. C. Seasonal and interannual assessment of cloud cover and atmospheric constituents across the Amazon (2000-2015): Insights for remote sensing and climate analysis. ISPRS Journal of Photogrammetry and Remote Sensing. *ISPRS Journal of Photogrammetry and Remote Sensing* , v. 145, p. 309-327, 2018.

Matsuoka, A.; Larouche, P.; Poulin, M.; Vinvent, W.; Hattori, H. Phytoplankton community adaptation to changing light levels in the southern Beaufort Sea, Canadian Arctic, Estuarine, Coastal and Shelf Science, 82, 537-546, 2009.

Matsuoka, A.; Bricaud, A., Bennerm R.; Para, J.; Sempéré, R.; Prieur, L.; Bélanger, S.; Babin, M. Tracing the transport of colored dissolved organic matter in water masses of the Southern Beaufort Sea: relationship with hydrographic characteristics, Biogeosciences, 9, 925–940, 2012.

Moreira - Turcq, P., Seyler, P., Guyot, JL e Etcheber, H. Exportation of organic carbon from the Amazon River and its main tributaries. Hydrol. Process., 17, 1329-1344, 2003

Peacock, M.; Evans, C. D.; Fenenr, N.; Freeman, C.; Gough, R.; Jones, T. G.; Lebron, I. UV-visible absorbance spectroscopy as a proxy for peatland dissolved organic carbon (DOC) quantity and quality: considerations on wavelength and absorbance degradation, *Environmental Science: Processes Impacts*, **16**, 1445-1461, 2014.

Sander de Carvalho, L. A. ; Faria Barbosa, C. C. ; LEÃO MORAES NOVO, E. M. ; de MORAES RUDORFF, C. . Implications of scatter corrections for absorption measurements on optical closure of Amazon floodplain lakes using the Spectral Absorption and Attenuation Meter (AC-S-WETLabs). Remote Sensing of Environment, v. 157, p. 123-137, 2015

Seekell, D. A., Lapierre, J.-F., & Cheruvelil, K. S. (2018). A geography of lake carbon cycling. Limnology and Oceanography Letters, 3(3), 49–56.

Shen, Y., Fichot, C. G., Benner, R. Floodplain influence on dissolved organic matter composition and export from the 400 Mississippi-Atchafalaya River system to the Gulf of Mexico. Limnology and Oceanography, 57 (4), 1149-1160, 2012.

Spencer, R. G., Aiken, G. R., Wickland, K. P., Striegl, R. G., Hernes, P. J. Seasonal and spatial variability in dissolved organic matter quantity and composition from the Yukon River basin, Alaska. Global Biogeochemical Cycles, 22 (4), 2008.

Toming, K., Kutser, T., Laas, A., Sepp, M., Paavel, B., Noges, T. First experiences in mapping lake water quality parameters with Sentinel-2 MSI imagery. Remote Sensing, 8 (8), 1–14, 2016.

Van Stan, J. T., & Stubbins, A. Tree-DOM: Dissolved organic matter in throughfall and stemflow. Limnology and Oceanography Letters, 3(3), 199–214, 2018.

Vantrepotte, V., Danhiez, F. P., Loisel, H., Ouillon, S., Mériaux, X., Cauvin, A., Dessailly, D. CDOM-DOC relationship in contrasted coastal waters: implication for doc retrieval from ocean color remote sensing observation. Optics Express, 23 (1), 33. https://doi.org/10.1364/oe.23.000033, 2015.

Volpato, G. L., Duarte, C. R. A., Luchiari, A. C. Environmental color affects Nile tilapia reproduction. Brazilian Journal of 415 Medical and Biological Research, 37, 479-483, 2004.